# Class-A penicillin binding proteins do not contribute to cell shape but repair cell-wall defects

Antoine Vigouroux[1,2,3†], Baptiste Cordier[1], Andrey Aristov[1], Laura Alvarez[4], Gizem Özbaykal[1,5], Thibault Chaze[6], Enno Rainer Oldewurtel[1], Mariette Matondo[6], Felipe Cava[4], David Bikard[2], Sven van Teeffelen[1]*

[1]Microbial Morphogenesis and Growth Laboratory, Institut Pasteur, Paris, France; [2]Synthetic Biology Laboratory, Institut Pasteur, Paris, France; [3]Université Paris Descartes, Sorbonne-Paris-Cité, Paris, France; [4]Laboratory for Molecular Infection Medicine Sweden (MIMS), Umeå Centre for Microbial Research (UCMR), Department of Molecular Biology, Umeå University, Umeå, Sweden; [5]Université Paris Diderot, Sorbonne-Paris-Cité, Paris, France; [6]Proteomics Platform, Institut Pasteur, Paris, France

*For correspondence:
sven.vanteeffelen@gmail.com

Present address: †Department of Microbiology, Institut Pasteur, Paris, France

Competing interests: The authors declare that no competing interests exist.

**Abstract** Cell shape and cell-envelope integrity of bacteria are determined by the peptidoglycan cell wall. In rod-shaped *Escherichia coli*, two conserved sets of machinery are essential for cell-wall insertion in the cylindrical part of the cell: the Rod complex and the class-A penicillin-binding proteins (aPBPs). While the Rod complex governs rod-like cell shape, aPBP function is less well understood. aPBPs were previously hypothesized to either work in concert with the Rod complex or to independently repair cell-wall defects. First, we demonstrate through modulation of enzyme levels that aPBPs do not contribute to rod-like cell shape but are required for mechanical stability, supporting their independent activity. By combining measurements of cell-wall stiffness, cell-wall insertion, and PBP1b motion at the single-molecule level, we then present evidence that PBP1b, the major aPBP, contributes to cell-wall integrity by repairing cell wall defects.

## Introduction

The peptidoglycan cell wall is responsible for both cell shape and mechanical integrity of the bacterial cell envelope (*Typas et al., 2010*; *Vollmer and Bertsche, 2008*). In Gram-negative bacteria such as *E. coli* the cell wall is a thin two-dimensional polymer that consists of mostly parallel glycan strands oriented circumferentially around the cell axis (*Gan et al., 2008*) and peptide cross-links that connect adjacent glycan strands. To avoid the formation of large pores in the cell wall during growth, cell-wall insertion and cell-wall cleavage must be tightly coordinated (*Vollmer et al., 2008*).

Cell-wall insertion involves two kinds of enzymatic reactions: transglycosylase (TGase) activity to extend the glycan strands, and transpeptidase (TPase) activity to create cross-links between glycan strands. During side-wall elongation, these two activities are carried out by two sets of machinery (*Cho et al., 2016*). First, the Rod complex comprises the Penicillin-Binding Protein PBP2, an essential TPase, and RodA, an essential TGase from the SEDS (shape, elongation, division and sporulation) family of proteins (*Meeske et al., 2016*; *Emami et al., 2017*). Together with the MreB cytoskeleton these and other Rod-complex components persistently rotate around the cell (*van Teeffelen et al., 2011*; *Dominguez-Escobar et al., 2011*; *Garner et al., 2011*; *Cho et al., 2016*; *Morgenstein et al., 2017*) and are responsible for rod-like cell shape. Second, bi-functional and essential class-A PBPs (aPBP's) PBP1a and PBP1b carry out both TPase and TGase activities. PBP1a and PBP1b are activated by the outer-membrane lipoprotein cofactors LpoA and LpoB, respectively (*Typas et al.,*

2010; *Paradis-Bleau et al., 2010*; *Typas et al., 2012*). Mutants in either PBP1a-LpoA or PBP1b-LpoB are viable and don't show any strong phenotype during regular growth, but mutants in components from both pairs are synthetically lethal (*Yousif et al., 1985*; *Typas et al., 2010*; *Paradis-Bleau et al., 2010*). aPBPs also interact with cell-wall cleaving lytic transglycosylases and DD-endopeptidases (*Banzhaf et al., 2020*), consistent with the possibility that they form multi-enzyme complexes responsible for both cell-wall expansion and insertion.

In the past, aPBPs have been suggested to work in close association with the MreB-based Rod complex (*Pazos et al., 2017*), motivated by biochemical interactions between PBP1a and the Rod-complex TPase PBP2 (*Banzhaf et al., 2012*), and by similar interactions between PBP1b and the divisome TPase PBP3 (*Bertsche et al., 2006*). However, each set of enzymes remains active upon inhibition of the respective other one and aPBPs and Rod-complex components show different subcellular motion (*Cho et al., 2016*). Furthermore, cells inhibited in PBP1ab activity lyse rapidly (*García del Portillo et al., 1989*; *Wientjes and Nanninga, 1991*), while cells inhibited in Rod-complex activity become round but do not immediately lyse (*Lee et al., 2014*).

Since aPBPs and Lpos form envelope-spanning complexes (*Egan et al., 2014*; *Jean et al., 2014*) they have been suggested to work as repair enzymes that activate at sites of defects or large pores in the cell wall (*Typas et al., 2012*; *Cho et al., 2016*). In support of this idea, aPBP activity is increased upon over-expression of the DD-endopeptidase MepS (*Lai et al., 2017*), which cleaves peptide bonds (*Singh et al., 2012*). Therefore, Rod complex and aPBPs might serve different functions despite catalyzing the same chemical reactions (*Zhao et al., 2017*; *Pazos et al., 2017*). In agreement with this viewpoint, recent work in the gram-positive *Bacillus subtilis* showed that the two machineries have opposing actions on cell diameter and lead to either circumferentially organized or disordered cell-wall deposition (*Dion et al., 2019*).

Based on the selective interactions between PBP1a-PBP2 and PBP1b-PBP3 (*Banzhaf et al., 2012*; *Bertsche et al., 2006*), and based on a mild localization of PBP1b at the cell septum (*Bertsche et al., 2006*), PBP1a was suggested to be mostly involved in cell elongation and PBP1b in cell division. However, PBP1b also contributes to cell elongation, where it might have an even more important role than PBP1a under normal growth conditions: PBP1b is found throughout the cell envelope (*Bertsche et al., 2006*; *Paradis-Bleau et al., 2010*), strains lacking PBP1b have greater mechanical plasticity in the cylindrical part of the cell (*Auer et al., 2016*), their overall rate of peptidoglycan (PG) insertion is reduced (*Caparrós et al., 1994*), they are more sensitive to chemicals perturbing cell-wall insertion, including mecillinam, A22, and D-methionine (*García del Portillo and de Pedro, 1991*; *Nichols et al., 2011*; *Caparrós et al., 1992*), they are more sensitive with respect to outer-membrane assembly defects (*Morè et al., 2019*), and they cannot recover from spheroplasts (*Ranjit et al., 2017*).

Here, we studied the role of aPBPs for cell shape and cell-wall integrity. First, we measured viability and cell shape during steady-state growth at different protein levels. We found that aPBPs have no role in maintaining cell shape and are therefore not required for proper Rod-complex activity. On the contrary, we confirmed that aPBPs are essential for mechanical cell-wall integrity. Second, we investigated how the major aPBP PBP1b contributes to mechanical integrity, simply through a higher overall rate of peptidoglycan insertion (*Caparrós et al., 1994*), by constitutively stabilizing the cell wall, for example by inserting peptidoglycan in a more spatially homogeneous manner, or through active repair of local cell-wall damage, as previously suggested (*Typas et al., 2012*; *Lai et al., 2017*; *Cho et al., 2016*). We first measured mechanical stability and amount of peptidoglycan in cells with aPBP levels reduced three-fold. These cells show reduced cell-wall stiffness and integrity while maintaining a high rate of peptidoglycan insertion. Therefore, aPBPs apparently strengthen the cell wall independently of changes in insertion rate. Increased integrity could then come about either through constitutive PBP1b activity or through an adaptive repair mechanism (*Typas et al., 2012*). Using a combination of cell-wall perturbations and time-dependent expression of PBP1b, we found evidence that PBP1b senses and repairs cell-wall defects. As a complementary approach, we used single-molecule tracking of a GFP-PBP1b fusion. We found that the bound, non-diffusive fraction of PBP1b molecules decreases with increasing PBP1b or PBP1a levels and increases with LpoB levels, suggesting that PBP1b localizes in a need-based manner, which is facilitated through LpoB. Second, we effectively increased the average cell-wall pore size by transiently inhibiting cell-wall insertion during growth. We found that the bound fraction of PBP1b molecules increases shortly after drug treatment and remains high up to 20 min after washout, supporting that PBP1b molecules directly respond to

cell-wall damage. Together, our results support the hypothesis that PBP1b is responsible for maintaining the integrity and structural organization of peptidoglycan on a local scale by inserting peptidoglycan in a targeted manner. On the contrary, neither of the two aPBPs has a role in cell-shape maintenance.

## Results

### Class-A PBPs are dispensable for cell shape but required for cell-envelope integrity

To investigate the importance of aPBPs for cell shape and cell-wall integrity, we constructed a strain with tunable levels of PBP1a and 1b using partial CRISPR knock-down, which reduces the transcription rate by a fractional amount (*Vigouroux et al., 2018*). To that end, we used the strain LC69 ($P_{tet}$-dCas9) (*Cui et al., 2018*) and fused PBP1a and PBP1b to RFP (mCherry variant) and GFP (sfGFP variant) respectively, in their native loci (strain AV44). We then used combinations of different CRISPR guides targeting GFP and RFP with a variable number of mismatches (*Vigouroux et al., 2018*). To extend the range of possible repression levels, CRISPR guides were expressed in two different forms: i) as a CRISPR RNA (crRNA) co-expressed with the tracrRNA, on the pCRRNAcos vector (*Vigouroux et al., 2018*), or ii) as a single-guide (sgRNA) with fused crRNA and tracrRNA, on the pAV20 vector (*Dion et al., 2019*) (*Figure 1A*). The CRISPR guides are named according to their complementarity to GFP (G) or RFP (R), Ø designating a control guide. Increasing complementary leads to increased repression (*Figure 1—figure supplement 1* and *Supplementary file 1*).

To ensure that no truncated or non-fluorescent form of PBP1a or PBP1b was produced, we used bocillin-labeled SDS-PAGE (*Figure 1—figure supplement 2A*). We quantified PBP1ab protein levels by combining relative mass spectrometry (Data-Independent Acquisition or DIA), absolute mass spectrometry (Parallel Reaction Monitoring or PRM), SDS-PAGE, and single-cell fluorescence measurements (see Materials and methods, *Figure 1—figure supplement 3*, *Table 1*).

The absolute number of PBP1b molecules per cell in the WT is 166 ± 26, in agreement with previous measurements (*Dougherty et al., 1996*). However, in AV44, levels of non-repressed RFP-PBP1a and GFP-PBP1b are 1300% and 370% higher than their homologs in the wild-type (*Table 1*), reminiscent of previous reports of elevated levels for fluorescent fusions (*Paradis-Bleau et al., 2010*). While not anticipated, this allowed us to explore aPBP levels ranging from strong repression to strong over-expression.

Interestingly, when repressing GFP-PBP1b, the residual expression was higher than what the same CRISPR guides would produce on constitutive GFP (*Figure 1—figure supplement 4*, left), suggesting that a form of negative feedback raises PBP1b expression in response to repression. We did not detect such a feedback for RFP-PBP1a (*Figure 1—figure supplement 4*, right), and we found only minor cross-talk between the levels of PBP1a and PBP1b at low PBP1b levels (*Table 1*).

As expected from the synthetic lethality of PBP1ab, strains with a strong repression of both PBP1a and PBP1b did not survive. In particular, in strain AV51 (ΔPBP1a), using the perfect-match sgRNA G20 which produces a repression to about 4% of the native level in AV44 (*Table 1*) leads to cell death. In contrast, AV51 with PBP1b repressed to 30% using sgRNA G14 (DIA, *Table 1*) is still viable. For all the strains that survive repression, the growth rate is unaffected, regardless of aPBP levels (*Figure 1B*).

To systematically measure the impact of PBP1ab levels on cell morphology, we varied the level of each PBP between 30% and 1300% (DIA) in strains lacking the respective other PBP. Expression level has hardly any effect on cell shape (*Figure 1C*). From lowest to highest PBP1a or PBP1b levels cell diameter increases by only 75 nm. In contrast, a 10-fold decrease in the level of Rod-complex-related operons PBP2-RodA or MreBCD increases diameter by about 800 nm (*Figure 1D*), as previously demonstrated (*Vigouroux et al., 2018*). Our observations are also in stark contrast to *B. subtilis*, where a similar change of the level of the major class-A PBP PBP1 leads to a 600 nm increase in diameter (*Dion et al., 2019*). As a control, we used an alternative setup based on the inducible $P_{ara}$ promoter (strains AV100, AV101) and checked the lack of any major shape phenotype at low PBP1ab induction (*Figure 1—figure supplement 2BC*).

We also examined the shape of cells that were depleted for PBP1ab to the point of lysis. To that end, we used time-lapse microscopy after induction of our strongest sgRNAs (pAV20 G20-R20,

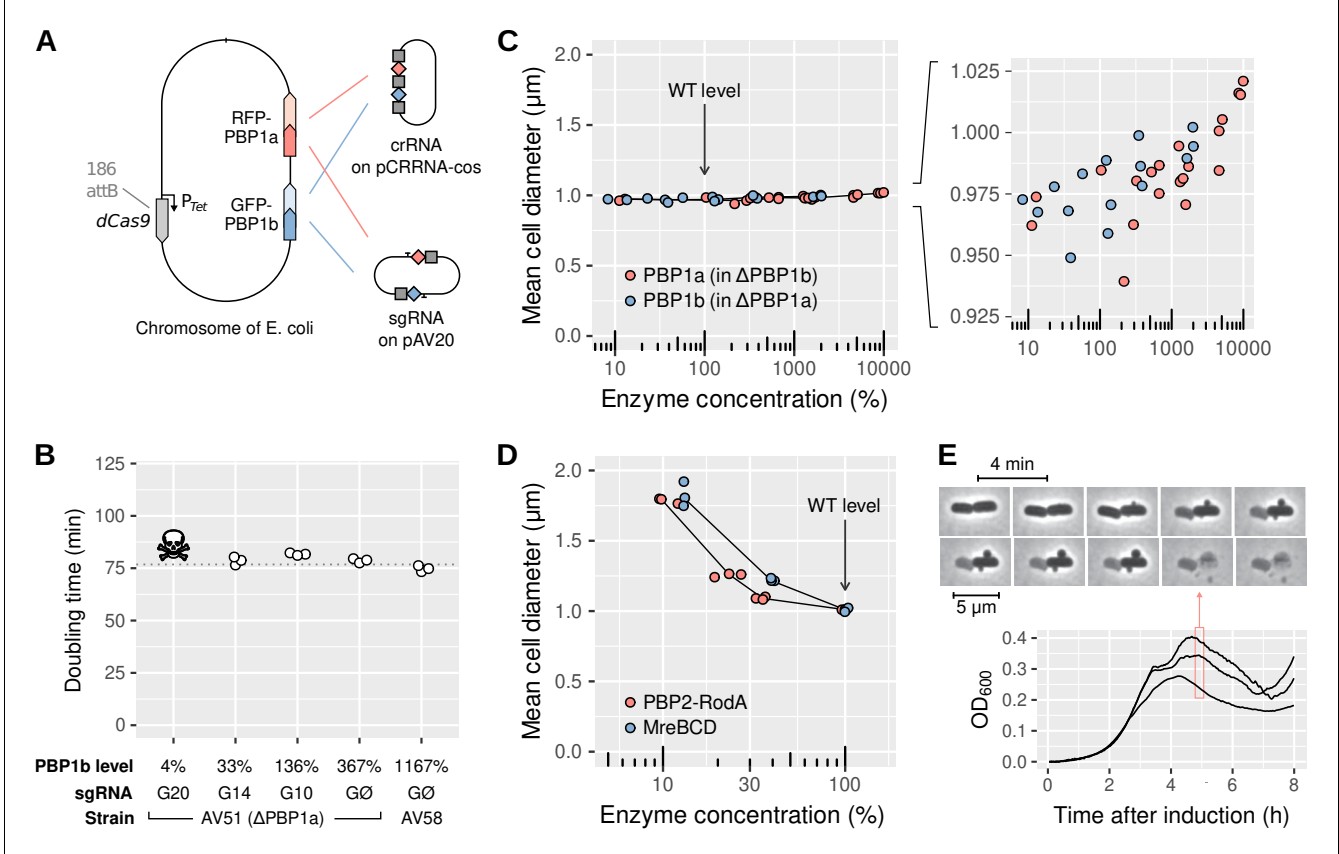

**Figure 1.** aPBPs have no role in maintaining rod-like cell shape. (**A**) Sketch of the strain AV44 (LC69 *mrcB::gfp-mrcB*, *mrcA::rfp-mrcA*) with tunable levels of RFP-PBP1a and GFP-PBP1b. CRISPR guides are expressed either as crRNA (top) or as sgRNA (bottom), see also *Figure 1—figure supplement 1*. (**B**) Doubling time of AV51 (AV44 ΔPBP1a)/pAV20 as a function of PBP1b level, in minimal medium with glucose and casamino-acids at 30°C. sgRNA are expressed from pAV20 as annotated. AV58 is AV51 P_ara_-GFP-PBP1b for over-expression. Skull logo: not viable. (**C**) Effect of aPBP concentration on cell diameter. Points indicate the median diameter within each population. Green: AV51/pCRRNAcos with crRNA G20, G14, G10 and GØ, or AV58 (over-expression). Red: AV50 (AV44 ΔPBP1b)/pCRRNAcos with crRNA R20, R18, R11 and RØ. AV63 is AV50 HK022::P_ara_-RFP-PBP1a for over-expression. Levels were determined based on fluorescence and normalized with respect to WT according to DIA. (**D**) Effect of the concentration of Rod-complex proteins on cell diameter. Green: AV88 (LC69 MreB-GFP)/pAV20 with sgRNA G14, G10 or GØ. Red: AV08 (LC69 RFP-PBP2)/pAV20 with crRNA R20, R18, R11 or RØ. (**E**) Growth curve of AV44/pAV20 with PBP1ab repressed to lethal level (sgRNA G20-R20), and cell morphology during lysis. Individual points are biological replicates. OD: optical density. WT: wild-type.

The online version of this article includes the following video, source data, and figure supplement(s) for figure 1:

**Source data 1.** Data used to generate *Figure 1* and its supplements.
**Figure supplement 1.** Passage probability of the different CRISPR guides used in this study.
**Figure supplement 2.** The RFP-PBP1a and GFP-PBP1b fusions are the only forms of aPBPs present in AV44.
**Figure supplement 3.** Quantification of GFP-PBP1b by semi-quantitative SDS-PAGE.
**Figure supplement 4.** Residual PBP1a and PBP1b levels in response to CRISPR-based repression, measured by fluorescence microscopy.
**Figure supplement 5.** Dimensions of individual cells before lysis due to PBP1ab repression, in LB Strain: AV44/pAV20 G20-R20.
**Figure 1—video 1.** Video of a cell of AV44/pAV20 with sgRNA G20-R20, bulging then lysing.
https://elifesciences.org/articles/51998#fig1video1

*Table 1*). Cells abruptly lysed without changes in cell dimensions compared to the minimum viable expression level (*Figure 1E*, *Figure 1—figure supplement 5* and *Figure 1—video 1*). However, we often observed small bulges on the sides of the cells just before lysis. This behavior, previously also observed upon LpoAB depletion (*Typas et al., 2010*), is similar to the effect of beta-lactam antibiotics (*Chung et al., 2009*), suggesting that cells accumulate lethal cell-wall defects in the absence of PBP1ab.

Together, our observations suggest that aPBPs are required for cell-wall integrity at the local scale but dispensable for maintenance of rod-like cell shape.

**Table 1.** Levels of PBP1ab expressed from different cassettes, and repressed using different sgRNA or crRNA.

The levels are determined using either fluorescence microscopy, SDS-PAGE with fluorescence detection, or mass spectrometry (DIA: Data-Independent Acquisition or PRM: Parallel Reaction Monitoring.), as described in the Materials and methods. RFP-PBP1a is either non-repressed (AV44) or deleted (AV51 and AV58). GFP-PBP1b is either non-repressed (AV44) or deleted (AV50 and AV63). Levels relative to LC69 are obtained by multiplying the levels relative to AV44 by the levels obtained by DIA for AV44, with propagated error. Ø: Control guides producing no repression. n.d.: not determined.

**Relative and absolute quantification of PBP1b**

| Strain | Promoter | System | Guide | Fluorescence (% of AV44) | Fluorescence (% of LC69) | Dia (%) | SDS-PAGE (copy/cell) | PRM (copy/cell) |
|---|---|---|---|---|---|---|---|---|
| LC69 | Wild-type | | | n.d. | n.d. | 100 | n.d. | 166 ± 28 (100%) |
| AV44 | Native fusion | sgRNA | G20 | 1.0 ± 0.04 | 3.8 ± 0.4 | n.d. | n.d. | n.d. |
| AV44 | Native fusion | sgRNA | G14 | 6.6 ± 0.79 | 24 ± 2.9 | 27 ± 2 | 40 ± 5 | 46 (28%) |
| AV51 | Native fusion | sgRNA | G14 | n.d. | n.d. | 33 ± 3 | 67 ± 14 | 56 ± 7 (33%) |
| AV51 | Native fusion | crRNA | G20 | 4.1 ± 2.0 | 15 ± 7.6 | n.d. | n.d. | n.d. |
| AV51 | Native fusion | crRNA | G14 | 12 ± 3.1 | 44 ± 12 | n.d. | n.d. | n.d. |
| AV51 | Native fusion | crRNA | G10 | 36 ± 2.8 | 131 ± 15 | n.d. | n.d. | n.d. |
| AV51 | Native fusion | crRNA | GØ | 97 ± 13 | 356 ± 57 | n.d. | n.d. | n.d. |
| AV44 | Native fusion | crRNA | GØ | 100 ± 5.4 | 367 ± 38 | 367 ± 32 | 688 ± 115 | 547 ± 52 (330%) |
| AV58 | P$_{ara}$ | crRNA | GØ | 509 ± 57 | 1870 ± 265 | n.d. | n.d. | n.d. |

**Relative quantification of PBP1a**

| Strain | Promoter | System | Guide | Fluorescence (% of AV44) | Fluorescence (% of LC69) | Dia (%) |
|---|---|---|---|---|---|---|
| LC69 | Wild-type | | | n.d. | n.d. | 100 |
| AV44 | Native fusion | sgRNA | R20 | n.d. | n.d. | 20 ± 2 |
| AV50 | Native fusion | crRNA | R20 | 3 ± 4 | 43 ± 56 | n.d. |
| AV50 | Native fusion | crRNA | R18 | 21 ± 4 | 278 ± 139 | n.d. |
| AV50 | Native fusion | crRNA | R11 | 46 ± 6 | 620 ± 298 | n.d. |
| AV50 | Native fusion | crRNA | RØ | 100 ± 1 | 1337 ± 615 | n.d. |
| AV44 | Native fusion | crRNA | RØ | 100 ± 7 | 1337 ± 622 | 1337 ± 615 |
| AV63 | P$_{ara}$ | crRNA | R18 | 166 ± 14 | 1549 ± 786 | n.d. |
| AV63 | P$_{ara}$ | crRNA | R11 | 355 ± 22 | 4750 ± 2204 | n.d. |
| AV63 | P$_{ara}$ | crRNA | RØ | 691 ± 50 | 9243 ± 4304 | n.d. |

The online version of this article includes the following source data for Table 1:

Source data 1. Data used to generate *Table 1*.

## At low levels of aPBPs, cells insert as much peptidoglycan but show reduced mechanical integrity

It was previously reported that a ΔPBP1b strain inserts about 50% less peptidoglycan, while a ΔPBP1a strain maintains a WT insertion rate (*Caparrós et al., 1994*). Thus, we aimed to study whether aPBPs maintain cell-wall integrity simply due to an elevated rate of cell-wall insertion or by modulating the cell wall structurally, for example, through a more homogeneous distribution of peptidoglycan material (*Typas et al., 2012*).

We extracted cell-wall sacculi from strains with different repression levels of PBP1a and PBP1b, and measured composition and amount through UPLC-UV. When PBP1b is deleted (AV105) or repressed strongly (AV84/pAV20 G20-RØ, *Table 1*), the amount of peptidoglycan is reduced to about one half, even with a high level of PBP1a (*Figure 2A*). AV84 and AV105 are AV44 ΔlysA and AV51 ΔlysA, respectively, and we used sgRNAs for repression throughout this section. Intriguingly, when we reduced both PBP1a and PBP1b to about 30% of WT using sgRNA G14-R20 in AV84, cells inserted as much peptidoglycan as the non-repressed strain with 1300% PBP1a and 370% PBP1b (*Figure 2A*). Therefore, the amount of peptidoglycan per cell is independent of PBP1ab levels as long as PBP1b is expressed at a minimum between 5% and 30% of native levels. Furthermore, the decrease of peptidoglycan insertion upon strong PBP1b repression or deletion cannot be

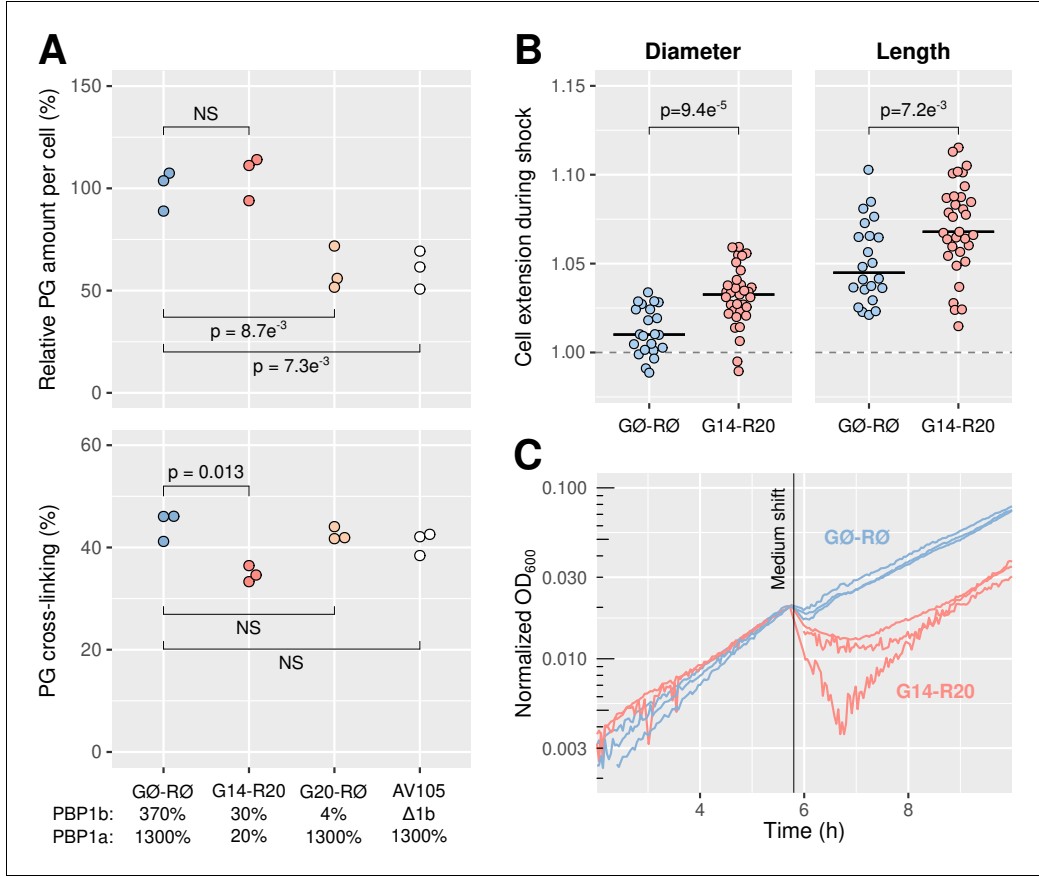

**Figure 2.** Repression of PBP1ab reduces mechanical stiffness while maintaining a high rate of peptidoglycan insertion. (**A**) Top: Steady-state amount of peptidoglycan per cell, measured in AV84 (AV44 ΔLysA)/pAV20 and AV105 (AV44 ΔPBP1b ΔLysA) as annotated. Bottom: Fraction of the peptidoglycan that is cross-linked in the same conditions. p-Values correspond to a two-sided t-test. NS: not significant. (**B**) Extension of the cells' short axis (left) and long axis (right) after a 1 osm/L NaCl downshock, in AV93 (AV44 ΔMscLS)/pAV20 with sgRNA GØ-RØ or G14-R20. A value of one corresponds to no extension. Horizontal lines represent the medians. p-Values correspond to a two-sided permutation test. (**C**) Growth curves before and after a 1 osm/L osmotic downshock, in AV93/pAV20 with sgRNA GØ-RØ or G14-R20. The curves are scaled so all curves have the same OD at the moment of the shock. OD: optical density.

The online version of this article includes the following video, source data, and figure supplement(s) for figure 2:

**Source data 1.** Data used to generate *Figure 2* and its supplements.

**Figure supplement 1.** Amount of incorporated $^3$H-mDAP per optical density as a function of time.

**Figure supplement 2.** UPLC-UV chromatograms of the peptidoglycan after digestion by muramidase.

**Figure supplement 3.** Growth curves before and after osmotic shock in minimal medium.

**Figure 2—video 1.** Video of sample cells in phase-contrast microscopy during an osmotic downshock.

https://elifesciences.org/articles/51998#fig2video1

compensated by high PBP1a expression, since a strain with 20% of PBP1a and 30% of PBP1b (AV84/pAV20 G14-R20) still inserts more peptidoglycan than a strain with 1300% PBP1a but no PBP1b (AV105/pAV20 GØ-RØ) (p=7.3 × $10^{-3}$, t-test). We obtained similar results when measuring the incorporation of the radiolabeled cell-wall precursor mDAP (meso-diaminopimelic acid) (*Wientjes et al., 1991*, *Figure 2—figure supplement 1*). Together, these data show that the total amount of peptidoglycan is rigorously buffered against variation in their levels, over a wide range of concentrations.

Looking closer at the chemical content of the peptidoglycan, we found that with PBP1a and PBP1b repressed to ≈30% (AV84/pAV20 G14-R20) the rate of cross-linking between glycan strands was reduced, from 45% to about 35% (p=0.013, t-test, *Figure 2A*, *Figure 2—figure supplement 2*

and *Supplementary file 2*). Thus, when PBP1ab levels are low, the amount of peptidoglycan per cell remains unchanged but the chemical content of the peptidoglycan is altered. Interestingly, in the absence of PBP1b, the cross-linking rate was similar to the non-repressed strain, despite the great reduction in peptidoglycan amount.

Previously, it has been reported that cells lacking PBP1b have a more elastic cell wall (*Auer et al., 2016*), as expected from their reduced peptidoglycan amount. Since a reduction in the levels of both PBP1a and PBP1b affects cell-wall chemical composition rather than its amount, we wondered if it also has an effect on the mechanical integrity of the cell wall.

To measure cell-wall elasticity, we submitted cells to an osmotic downshock of 1 osm/L of NaCl under the microscope, similarly to *Buda et al. (2016)* (*Figure 2B*). The sudden increase of turgor pressure causes an increase of cell dimensions that is inversely related to cell-wall stiffness, in agreement with *Buda et al. (2016)* (*Figure 2—video 1*). To avoid rapid response to osmotic shock, we deleted the mechano-sensitive channels *mscS* and *mscL* from AV44 (strain AV93). We found that repression of PBP1ab to about 30% of WT levels (AV93/pAV20 G14-R20) leads to a decrease of both axial and circumferential stiffness if compared to the non-repressed strain (*Figure 2B*). Therefore, the reduced number of PBP1ab is likely less capable to protect the cell wall against the accumulation of mechanical defects, despite unperturbed peptidoglycan density.

As a potential consequence of reduced mechanical integrity, we next studied cell survival after osmotic shock in batch culture (*Figure 2C* and *Figure 2—figure supplement 3*). In AV93 repressed for PBP1ab (pAV20 G14-R20), the osmotic shock causes death of a large fraction of cells, while the non-repressed cultures (pAV20 GØ-RØ) are mostly able to survive.

In summary, we found that cells with reduced levels of PBP1ab show reduced mechanical stiffness and integrity, which leads to an increased rate of cell death upon osmotic downshock, despite unperturbed levels of peptidoglycan amount. Therefore, the lack of PBP1ab affects cell-wall local structure independently of peptidoglycan density.

## PBP1b likely repairs cell-wall damage

PBP1ab could in principle increase mechanical integrity in two different ways: through constitutive peptidoglycan synthesis that merely compensates the accumulation of mechanical defects, or through a targeted repair mechanism that inserts cell wall in response to damage, as previously proposed (*Typas et al., 2012*). To discriminate these two possibilities, we studied the ability of PBP1b to sustain and recover from transient inhibition of peptidoglycan insertion. We blocked the production of peptidoglycan precursors by treating cells with D-cycloserine, which inhibits L-alanine-to-D-alanine conversion and D-alanine-D-alanine ligation (*Lambert and Neuhaus, 1972*).

Upon treatment with a high concentration of D-cycloserine (1 mM) under the microscope, cells continue to elongate at a nearly unperturbed rate for about 40–50 min before they suddenly lyse (*Figure 3A* and *Figure 3—video 1*). Notably, cell-wall synthesis is affected well before lysis according to the rotational motion of MreB (*Figure 3—figure supplement 1A*, *Figure 3—video 2*), which requires cell-wall insertion (*van Teeffelen et al., 2011*). Arrest of cell-wall insertion well before lysis was also demonstrated by independent [3]H-mDAP labeling (*Oldewurtel et al., 2019*). As cells continue to elongate while inserting material at a severely reduced rate, we reasoned that the density of the cell wall must decrease. In agreement, UPLC analysis of cell-wall sacculi shows that the fraction of cell-wall cross-links is reduced by about one third after 1 hr of 1 mM D-cycloserine treatment (*Figure 3B*, *Figure 3—figure supplement 2* and *Supplementary file 2*). Decreased cross-linking likely causes a concomitant accumulation of defects, that is locations with increased pore size, which are ultimately responsible for lysis. Since UPLC-experiments were performed in rich growth medium (LB), we also checked that D-cycloserine treatment in LB had a similar behavior on cell lysis (*Figure 3—figure supplement 3*) and on MreB rotation (*Figure 3—figure supplement 1C*) as in minimal medium.

We then expressed PBP1b from a multi-copy plasmid (pBC03) under the control of an inducible $P_{ara}$ promoter in a ΔPBP1b background for rapid and wide modulation of PBP1b levels. In batch-culture experiments, WT cells, PBP1b-induced cells (PBP1b+), and non-induced cells (PBP1b-) lyse almost at the same time on average (*Figure 3C*), demonstrating that the structure of the cell wall prior to drug treatment and the presence of PBP1b during drug treatment have no impact on cell survival. We observed the same behavior in LB (*Figure 3—figure supplement 3*).

To study the potential role of PBP1b for cell-wall repair, we washed out D-cycloserine right before rapid lysis would have started (after 32 min in minimal media or after 22 min in LB), and monitored growth (*Figure 3D*, *Figure 3—figure supplement 3B*). We observed that optical density ($OD_{600}$) dropped for PBP1b- cells, while no drop occurred for PBP1b+ cells. We thus reasoned that during an extended period of about 1 hr the rate of cell lysis was higher in PBP1b- cells than in PBP1b+ cells. To test this hypothesis, we conducted single-cell timelapses of PBP1b+ and PBP1b- cells and monitored growth and lysis directly (*Figure 3E–G*). Consistent with the $OD_{600}$ measurements, about half of the PBP1b- but almost none of the PBP1b+ cells lysed between 10–25 min after drug washout according to manual analysis (*Figure 3—figure supplement 4*).

To discriminate whether the elevated rate of recovery is due to increased mechanical integrity prior to drug treatment or due to PBP1b activity after washout, we induced PBP1b expression only 5–10 min before washout (*Figure 3D*, *Figure 3—figure supplement 3B*). We found that these cells recover nearly as well as cells expressing PBP1b during the whole experiment, according to $OD_{600}$ curve and single-cell analysis (*Figure 3*). The alleviating effect of PBP1b takes place in less than 20 min after drug removal in both minimal medium and LB (*Figure 3—figure supplement 5*). In minimal medium, this time constitutes less than 30% of the generation time (*Figure 3F*). On the contrary, PBP1b- cells recover only after about one generation time. Thus, PBP1b is likely recruited to local cell-wall defects, where it then serves as a repair enzyme.

It was previously suggested that PBP1b together with LpoB effectively serve as sensors of large cell-wall pores (*Typas et al., 2012*). To test this hypothesis, we used PBP1b(E313D) (hereafter called PBP1b*), a mutant that is locked in the active conformation and inserts cell wall in the absence of LpoB (*Markovski et al., 2016*). Using CRISPR repression in strain AV130 (GFP-PBP1b* and RFP-PBP1a in native locus, ΔLpoB), we confirmed that GFP-PBP1b* alone can fulfill the essential role of aPBPs for exponential growth (*Figure 3—figure supplement 6A*). Then, we transiently treated strain AV128 (GFP-PBP1b*, ΔLpoB) with D-cycloserine as above. We found that GFP-PBP1b* recovered better than ΔPBP1b but worse than the non-mutated, LpoB-activated GFP-PBP1b, with a delay of recovery of about 40 min (*Figure 3—figure supplement 6B*). Therefore, the interaction with LpoB is necessary for immediate recovery from damage, even though the constitutive activity of the mutant in ΔLpoB is sufficient to maintain exponential growth. Our findings therefore support the hypothesis that PBP1b together with LpoB is able to detect and repair regions in need for cell-wall insertion.

## PBP1b efficiently prevents lysis during reduced precursor availability

As a complementary approach to D-cycloserine treatment, we also inhibited peptidoglycan insertion by transiently starving an auxotrophic mutant (*asd-1*) for the essential peptidoglycan component mDAP (*Hatfield et al., 1969*, *Figure 3—figure supplement 3C–D*). We observed similar but slightly different behavior as for D-cycloserine treatment. In LB, mDAP depletion induces lysis after about 60 min on average in PBP1b+ cells, suggesting that cell-wall synthesis is inhibited later than during D-cycloserine treatment, in agreement with previous experiments (*van Teeffelen et al., 2011*). Here, we observed that MreB-GFP rotation is severely reduced within 10–20 min (*Figure 3—figure supplement 1D* and *Figure 3—video 2*), a similar time as after D-cycloserine treatment. Possibly, a low, non-detected level of cell-wall synthesis is still ongoing. In contrast to PBP1b+ cells, PBP1b- cells start to lyse after about 35 min (*Figure 3—figure supplement 3C*). We reasoned that PBP1b might prevent or repair damage efficiently already during mDAP depletion, with a reduced amount of cell-wall precursors available.

After re-addition of mDAP following 32 min of mDAP depletion we found that expressing PBP1b right before mDAP repletion has an immediate effect on survival, leading to improved recovery compared to PBP1b- (*Figure 3F*). However, PBP1b+ recovers even faster, presumably because PBP1b also helps maintain cell-wall integrity during the 32 min of mDAP depletion.

At a sub-lethal concentration of D-cycloserine (100 µM), ΔPBP1b cells lyse after about 60 min, while the WT continues to grow, as reported previously (*Nichols et al., 2011*; *Figure 3—figure supplement 7*). Similarly to the mDAP depletion experiment, PBP1b+ cells presumably have the capacity to use the reduced pool of peptidoglycan precursors to counter the accumulation of mechanical defects. This is only the case with the wild-type, LpoB-activated PBP1b. With the PBP1b* mutant in a ΔLpoB background, the population collapses like a ΔPBP1b strain, further supporting that LpoB aids efficient defect repair.

Together, our findings suggest that PBP1b responds to cell-wall damage by repairing defects.

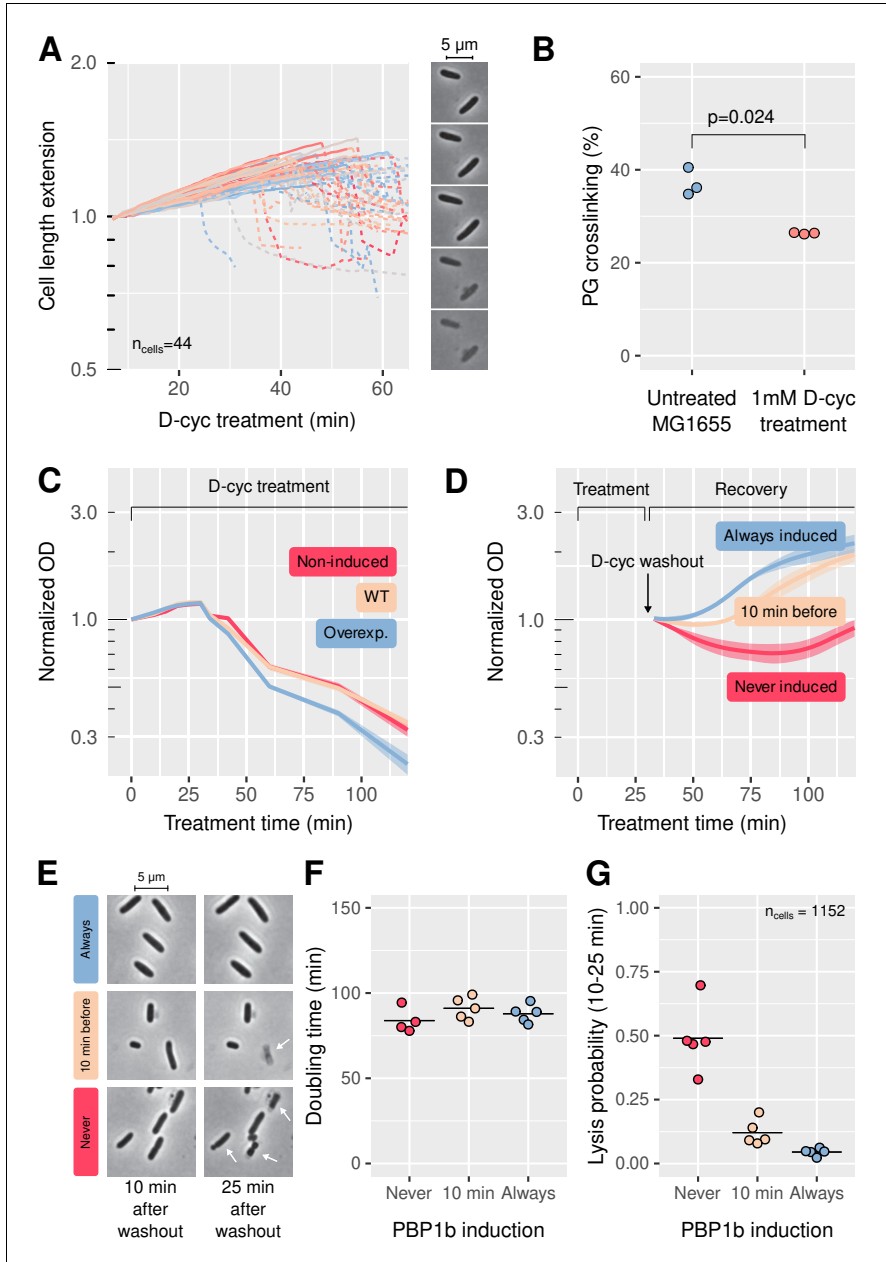

**Figure 3.** PBP1b facilitates quick recovery from transient inhibition of peptidoglycan synthesis. (**A**) Cell elongation before lysis from D-cycloserine treatment (1 mM) under the microscope, including sample snapshots. Strain is MG1655. Length is normalized to the length at the beginning of the movie. Solid lines describe growing cells, dashed lines correspond to phase-bright, lysing cells. Colors are arbitrary. (**B**) Effect of 1 mM D-cycloserine treatment on the cell-wall cross-linking rate, measured by UPLC on MG1655 before and after 1 hr of treatment. (**C**) Sensitivity to 1 mM D-cycloserine, for MG1655 (WT) and B150 (ΔPBP1b)/pBC03 (pBAD33-$P_{ara}$PBP1b) with arabinose (overexp.) or without arabinose (non-induced). (**D**) Recovery after washout from 32 min of D-cycloserine treatment (1 mM), for B150/pBC03. PBP1b is either always induced, never induced, or induced 10 min before D-cycloserine washout. In (**C-D**), shaded areas correspond to mean ± standard deviation of three biological replicates. (**E**) Sample images of B150/pBC03 taken 10 min and 25 min after washout from 1 mM D-cycloserine. PBP1b is either induced from the beginning, induced 10 min before washout, or never induced. White arrows point to cells that died during the 15 min imaging window. (**F**) Doubling time measured from single living cells, over the 15 min imaging window, for the three different PBP1b induction times. Each point represents a field of view with 80 ± 40 cells (total 1152). (**G**) Fraction of the cells that visibly lysed during the 15 min imaging window. WT: wild-type. Normalized OD: optical density normalized to the initial value, at the beginning of the treatment or recovery. All panels are done in M63 minimal medium, except cultures on panel B that were prepared in LB.

*Figure 3 continued on next page*

*Figure 3 continued*

The online version of this article includes the following video, source data, and figure supplement(s) for figure 3:

**Source data 1.** Data used to generate *Figure 3* and its supplements.

**Figure supplement 1.** Effect of depletion of peptidoglycan precursors measured by MreB motion.

**Figure supplement 2.** UPLC-UV analysis of the peptidoglycan, either on cells grown without treatment, or after 1 hr of 1 mM D-cycloserine treatment in MG1655 in LB.

**Figure supplement 3.** Damage sensitivity and recovery in LB.

**Figure supplement 4.** Detection of lysis events during D-cycloserine recovery.

**Figure supplement 5.** Alleviating effect of PBP1b expression after transient D-cycloserine treatment.

**Figure supplement 6.** The E313D mutant of PBP1b (PBP1b*) enables growth without LpoB, but does not rescue D-cycloserine recovery.

**Figure supplement 7.** Sensitivity to 100 µM D-cycloserine at different levels of PBP1b.

**Figure 3—video 1.** Movie of MG1655 cells lysing following D-cycloserine treatment in LB.
https://elifesciences.org/articles/51998#fig3video1

**Figure 3—video 2.** Sample of movies used for single-molecule tracking of MreB-GFP$_{SW}$, with overlaid trajectories.
https://elifesciences.org/articles/51998#fig3video2

## PBP1b immobilizes in response to cell-wall defects

To investigate the response of PBP1b to cell-wall damage at the molecular level, we studied the movement of individual GFP-PBP1b molecules in the inner membrane.

Previously, single-molecule tracking of PBP1a in *E. coli* (*Lee et al., 2016*) and PBP1 in *B. subtilis* (*Cho et al., 2016*) revealed that enzymes can be be divided in two populations: a diffusive fraction and a 'bound' fraction with near-zero diffusion coefficient. Presumably, only the bound fraction can insert peptidoglycan, while the diffusive fraction is searching for new insertion sites. Notably, bound molecules were detected for a duration of at most a few seconds, which did not allow to identify any persistent motion expected from processive transglycosylation.

To localize individual GFP-PBP1b molecules, we first photobleached a large fraction of them in HILO (highly inclined and laminated optical sheet) or epifluorescence mode and then tracked single GFP-PBP1b molecules with an imaging interval of 60 ms in HILO mode (*Figure 4—video 1*). The fraction of bound molecules was measured by fitting the observed distributions of single-molecule displacements to a two-state model using the Spot-On tool (*Hansen et al., 2018*, *Figure 4—figure supplement 1*).

In order to approximate WT levels in our fluorescently labeled strain, we used the crRNAs G10 and R18, which lead to an expression of about 130% for GFP-PBP1b and 280% for RFP-PBP1a in strains AV51 and AV50, respectively (*Table 1*). Using a ΔPBP1a background (AV51), we found about 20% of the PBP1b molecules to be bound around WT levels (crRNA G10), while 80% of the molecules moved diffusively with a diffusion constant of about 0.075 µm$^2$/s. Bound molecules were found all along the cell axis, and not only at mid-cell (*Figure 4—figure supplement 2*).

Qualitatively similar to the observations on the major aPBP PBP1 in *B. subtilis* (*Cho et al., 2016*), we found that the bound fraction of PBP1b decreases with increasing concentration (*Figure 4A*), suggesting that the activity of individual PBP1b enzymes is reduced upon increasing levels.

The activity of each PBP1b molecule could be limited by the availability of LpoB, the availability of peptidoglycan precursors, or the abundance of potential sites for cell-wall insertion, henceforth referred to simply as 'defects'. We therefore aimed to identify the potentially limiting factors by modulating protein levels and precursor availability. Over-expressing LpoB from plasmid pBC01 (pAM238-P$_{lac}$-*lpoB*) in AV44/pAV20 G10-R18 indeed increases the bound fraction, while deleting LpoB (strain AV110/pAV20 G10-R18) reduces the bound fraction (*Figure 4B*), indicating that the physical interaction with LpoB aids PBP1b immobilization or stabilizes the bound form. A lower bound fraction was also observed for the PBP1b* mutant, which is active even in a ΔLpoB background (*Figure 4—figure supplement 3*). Therefore, the reduced bound fraction is not merely due to reduced enzymatic activity.

Next, we investigated how PBP1b molecules respond to changes of PBP1a abundance. Maintaining PBP1b at about 30% of WT, we found that when PBP1a is deleted, the bound fraction of PBP1b is fourfold higher than during over-expression of PBP1a (1300% with respect to WT, *Figure 4C*). Since PBP1a and PBP1b do not share their outer-membrane activators LpoA and LpoB (*Typas et al.,*

*2010*), we reasoned that PBP1a affects PBP1b indirectly through its enzymatic activity. This could happen through depletion of the common precursor pool or by reducing the number of cell-wall defects detected by PBP1b-LpoB pairs.

To test whether PBP1b-immobilization depends on the availability of precursors or rather on the structure of the cell wall, we transiently inhibited precursor synthesis using D-cycloserine and thus increased the number of cell-wall defects as in *Figure 3C*.

We observed a rise of the bound fraction of PBP1b within less than 20 min and a subsequent increase to 37% within 40 min (*Figure 4D*), while the diffusion constant of diffusive PBP1b molecules is only mildly reduced (*Figure 4—figure supplement 4*). To make sure that the increase of the bound fraction is due to live, non-lysed cells, we investigated cell shape and also used the nucleic-acid dye propidium iodide, which only penetrates the membranes of dead cells. We then confirmed that visibly dead cells only contributed a small number of tracks to our dataset. Furthermore, while the number of lysed cells visibly rose during the latest set of movies corresponding to *Figure 4D* (from about 5% to 20% according to visual inspection), we did not observe a concurrent increase of the bound fraction. Our experiments therefore suggest that PBP1b molecules immediately respond to damage by increased binding. Furthermore, since this happens later than the arrest in MreB motion (a readout for the availability of precursors), it means that PBP1b immobilization does not require PBP1b activity.

We then also investigated whether PBP1b shows a higher bound fraction during recovery from D-cycloserine, where the presence of PBP1b greatly increases the chance of cell survival. In agreement with our expectation, we found that the bound fraction is elevated for about 20 min after a 30 min period of D-cycloserine treatment (*Figure 4E*). MreB motion is restored within the same time frame (*Figure 3—figure supplement 1B*).

In summary, our tracking results support the hypothesis that PBP1b together with its cognate activator LpoB contributes to cell-wall integrity by localizing and inserting peptidoglycan in response to local cell-wall defects.

## Discussion

In conclusion, our work suggests that different cell-wall-synthesizing machineries have distinct functions in *E. coli*. While the Rod complex is essential for rod shape, the bifunctional aPBPs PBP1ab have hardly any effect on cell shape, up to the point of cell lysis. Therefore, class-A PBPs are likely not required for the shape-preserving activity of the Rod complex. However, PBP1ab are essential for mechanical cell-wall integrity, and our experiments suggest that PBP1b repairs cell-wall defects by inserting peptidoglycan in response to local cell-wall defects, as previously hypothesized (*Cho et al., 2016*; *Typas et al., 2012*). This study therefore contributes to a growing body of evidence suggesting that the local mechanical and structural state of the cell wall provides a major physical cue for peptidoglycan remodeling and insertion.

While cell shape is hardly affected by PBP1ab repression, we found a mild but significant positive correlation between cell diameter and PBP1ab levels, consistent with a previous study of a ΔPBP1a mutant (*Banzhaf et al., 2012*). A much stronger correlation of same sign between PBP1 level and cell diameter was recently also observed in *B. subtilis* (*Dion et al., 2019*). In both species, it is conceivable that increased aPBP activity depletes a common pool of lipid II precursors and thus indirectly reduces the capacity of the Rod complex to maintain a narrower cell diameter. What then is responsible for the qualitatively different effect of aPBP levels on cell diameter in *E. coli* and *B. subtilis*? First, our tracking experiments showed that the bound fraction of PBP1b molecules is negatively correlated with PBP1ab expression and positively correlated with LpoB levels. These findings support the model that PBP1b activity is controlled by both the structure of the cell-wall substrate and by the presence of LpoB. In *B. subtilis*, cell-wall synthetic activity of PBP1 might be less regulated, even if PBP1 molecules are more mobile at high expression level (*Cho et al., 2016*). Second, the flux of lipid-II precursors shared by both systems might not be fixed in *E. coli*. Instead, both systems might secure access somewhat independently, a hypothesis also supported by the overall increase in peptidoglycan synthesis upon MepS over-expression (*Lai et al., 2017*). Finally, LpoB might be an important limiting factor for PBP1b activity, which is absent in *B. subtilis*.

At first sight, our observation of an increasing PBP1b-bound fraction with decreasing PBP1a levels seems to be in contradiction to previous measurements of PBP1b diffusion (*Lee et al., 2016*). Lee

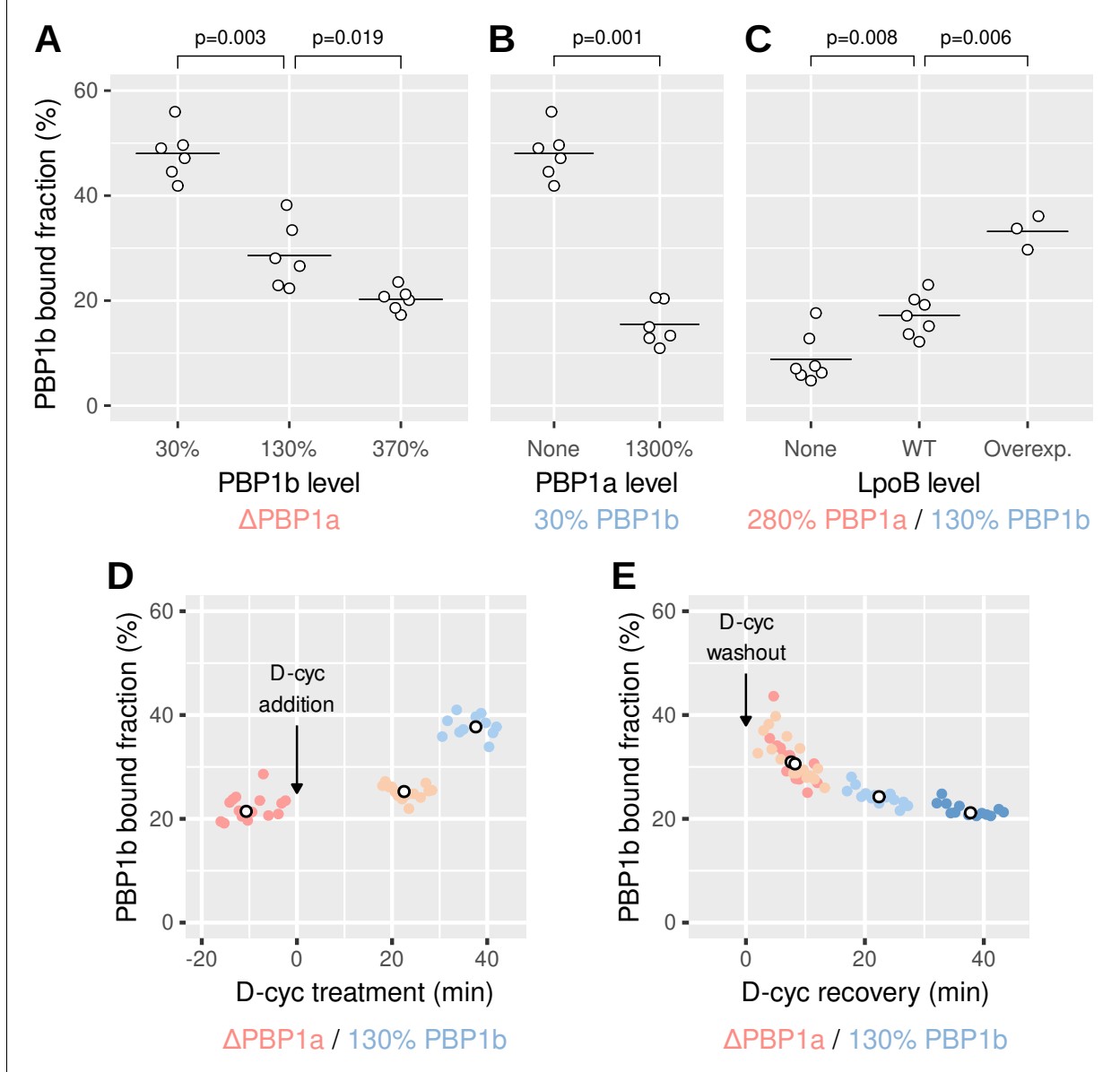

**Figure 4.** PBP1b localizes depending on the need for peptidoglycan synthesis. (**A-C**) Calculated bound fraction of PBP1b at different levels of PBP1b, PBP1a and LpoB, using strains AV44, AV51 (ΔPBP1a) or AV110 (ΔLpoB). For GFP-PBP1b, sgRNA G14 (in pAV20), crRNA G10 (in pCRRNAcos) and crRNA GØ (in pCRRNAcos) are used to reach 30%, 130% and 370%, respectively. For RFP-PBP1a, sgRNA R20 (in pAV20), crRNA R18 (in pCRRNAcos) and crRNA RØ (in pCRRNAcos) are used to reach 20%, 280% and 1300%, respectively. Each point represents a biological replicate comprising at least 5000 tracks. Horizontal lines are means. p-Values are from permutation tests. (**D-E**) Bound fraction of PBP1b at different times during 1 mM D-cycloserine treatment (**D**) and during recovery from 30 min of D-cycloserine treatment (**E**) in the strain AV51/pCRRNAcos G10-RØ. Colored points are individual movies and white points are medians from one sample. Corresponding free diffusion coefficients are shown in *Figure 4—figure supplement 4*. The online version of this article includes the following video, source data, and figure supplement(s) for figure 4:

**Source data 1.** Data used to generate *Figure 4* and its supplements.
**Figure supplement 1.** Tracking of single molecules and quantification of the fraction of bound molecules.
**Figure supplement 2.** Localization of bound molecules.
**Figure supplement 3.** Calculated bound fraction of the PBP1b* mutant, compared to PBP1b with and without LpoB.
**Figure supplement 4.** Diffusion coefficient of non-bound molecules during D-cycloserine treatment or recovery.
**Figure 4—video 1.** Sample of TIRF-microscopy movies used for single-molecule tracking of GFP-PBP1b, with overlaid trajectories.
https://elifesciences.org/articles/51998#fig4video1

et al. reported that deletions of either LpoB or PBP1a hardly affected the average diffusion constant of PBP1b molecules. Similar to our approach, they fused PBP1b to a fluorescent protein (PAm-Cherry) in the native chromosomal locus. Therefore, it is possible that PBP1b expression was also elevated in their strain, similar to our GFP-PBP1b fusion. We found that at high levels the bound fraction of PBP1b is low, even in the absence of PBP1a (*Figure 4A*). Expressing PBP1a might then only elicit a small relative change of the PBP1b bound fraction that is hard to detect in the average diffusion constant.

Consistently with average diffusion constants reported in *Lee et al. (2016)* we found that upon LpoB deletion the bound fraction of PBP1b is significantly reduced but does not completely disappear. This observation suggests that PBP1b does not strictly require LpoB for binding. We reasoned that PBP1b might be able to autonomously detect defects or sites for cell-wall insertion in the absence of LpoB. This hypothesis is consistent with the previous identification of a PBP1b* mutant that suppresses the lethality of a ΔPBP1a, ΔLpoB background (*Markovski et al., 2016*) and with the high residual activity of PBP1b in the absence of LpoB in vitro (*Paradis-Bleau et al., 2010*). However, while LpoB does not seem to be required for binding and PBP1b* activity during regular growth, we found that LpoB is required for rapid recovery from transient cell-wall-synthesis arrest (*Figure 3—figure supplement 6*). We therefore speculate that LpoB improves the propensity of PBP1b (or PBP1b*) to recognizes regions of the cell wall requiring repair.

As an alternative hypothesis, PBP1b might also immobilize independently of the cell wall in the absence of LpoB, through association with one or multiple different proteins that immobilize independently of LpoB. For example, it has been suggested that aPBPs interact with hydrolytic enzymes and with an outer-membrane bound nucleator of cell-wall hydrolases (*Banzhaf et al., 2020*). As another possibility, a fraction of PBP1b molecules might co-localize with the Rod complex or the divisome. We have recently shown that Rod-complex activity remains surprisingly high upon RodA depletion (*Wollrab et al., 2019*). Potentially, a different transglycosylase might compensate for the absence of RodA. It will thus be interesting to study the possibility of PBP1b or PBP1a to rescue Rod-complex activity in the absence of RodA in the future.

The precise nature of the defects targeted by PBP1b is still unknown. However, the fact that such defects accumulate when PG precursors become unavailable suggests that they are being generated during cell-wall enlargement, when cell-wall hydrolases and other enzymes cleave existing material. An attractive hypothesis previously suggested is that PBP1b and LpoB come into contact in a probabilistic fashion, depending on the cell-wall thickness and local pore size, independently of any particular structural feature (*Typas et al., 2012*). The function of class-A PBPs for constitutive growth could then be seen as homogenizing the cell wall, by directing cell-wall insertion to regions of lower density and defects.

In contrast to class-A PBPs, the Rod complex likely inserts cell wall in a more persistent manner (*Cho et al., 2016*). While Rod-complex initiation might depend on local cell-wall features (*Wollrab et al., 2019*), it seems likely that the Rod complex is less sensitive to cell-wall structure during processive activity.

While repair enzymes are well understood in the context of DNA damage, PBP1b is the first enzyme known to be involved in the repair of the peptidoglycan cell wall. Yet, the cell wall experiences nearly constant damage due to cell-wall expansion during growth or due to the action of cell-wall antibiotics, making repair all the more important. Recent work by some of us demonstrates that cell-wall cleavage likely happens in regions of elevated mechanical strain and stress (*Wong et al., 2017*). In the absence of repair, increased hydrolytic activity in regions of increased strain would then rapidly lead to more strain and eventually to lysis, as also predicted by computational simulations (*Furchtgott et al., 2011*) and as observed upon depletion of PBP1ab (*Figure 1E*) or upon treatment with peptidoglycan-synthesis inhibitors (*Yao et al., 2012*). We therefore think that more enzymes might insert peptidoglycan in a manner dependent on the local structure of the cell wall. Consistently, we recently demonstrated that the Rod complex initiates at locations that are independent of MreB and possibly determined by the cell wall itself (*Wollrab et al., 2019*). In the future, the challenge remains to identify the particular local features of the cell wall that attract different cell-wall-modifying enzymes.

# Materials and methods

## Key resources table

| Reagent type (species) or resource | Designation | Source or reference | Identifiers | Additional information |
|---|---|---|---|---|
| Strain, strain background (*E. coli*) | LC69 | *Cui et al., 2018* | 186::*Ptet-dcas9* | |
| Strain, strain background (*E. coli*) | AV03 | *Vigouroux et al., 2018* | 186::*Ptet-dcas9, HK022::P, λ::P* | |
| Strain, strain background (*E. coli*) | AV04 | *Vigouroux et al., 2018* | 186::*Ptet-dcas9, λ::P-mcherry* | |
| Strain, strain background (*E. coli*) | AV08 | *Vigouroux et al., 2018* | 186::*Ptet-dcas9, mrdA::mcherry-mrdA* | |
| Strain, strain background (*E. coli*) | AV29 | This work | 186::*Ptet-dcas9, ΔmrcB* | *Supplementary file 1* |
| Strain, strain background (*E. coli*) | AV31 | This work | 186::*Ptet-dcas9, mrcB::msfgfp-mrcB* | *Supplementary file 1* |
| Strain, strain background (*E. coli*) | AV44 | This work | 186::*Ptet-dcas9, mrcB::msfgfp-mrcB, mrcA::mcherry-mrcA* | *Supplementary file 1* |
| Strain, strain background (*E. coli*) | AV47 | This work | 186::*Ptet-dcas9, HK022::P-msfgfp, λ::P-mcherry* | *Supplementary file 1* |
| Strain, strain background (*E. coli*) | AV50 | This work | 186::*Ptet-dcas9, mrcA::mcherry-mrcA, ΔmrcB* | *Supplementary file 1* |
| Strain, strain background (*E. coli*) | AV51 | This work | 186::*Ptet-dcas9, mrcB::msfgfp-mrcB, ΔmrcA* | *Supplementary file 1* |
| Strain, strain background (*E. coli*) | AV58 | This work | 186::*Ptet-dcas9, mrcB::msfgfp-mrcB, ΔmrcA, HK022::Para-msfgfp-mrcB* | *Supplementary file 1* |
| Strain, strain background (*E. coli*) | AV63 | This work | 186::*Ptet-dcas9, mrcA::mcherry-mrcA, ΔmrcB, HK022::Para-mCherry-mrcA* | *Supplementary file 1* |
| Strain, strain background (*E. coli*) | AV67 | This work | 186::*Ptet-dcas9, mrcB::msfgfp-mrcB, mrcA::mcherry-mrcA, ΔpbpC* | *Supplementary file 1* |
| Strain, strain background (*E. coli*) | AV80 | This work | 186::*Ptet-dcas9, mrcB::msfgfp-mrcB, mrcA::mcherry-mrcA, ΔpbpC, ΔmtgA* | *Supplementary file 1* |
| Strain, strain background (*E. coli*) | AV84 | This work | 186::*Ptet-dcas9, mrcB::msfgfp-mrcB, mrcA::mcherry-mrcA, ΔpbpC, ΔmtgA, ΔlysA* | *Supplementary file 1* |

*Continued on next page*

*Continued*

| Reagent type (species) or resource | Designation | Source or reference | Identifiers | Additional information |
|---|---|---|---|---|
| Strain, strain background (*E. coli*) | AV88 | *Dion et al., 2019* | 186::*Ptet-dcas9, mreB::mreB-msfGFP* | |
| Strain, strain background (*E. coli*) | AV92 | This work | 186::*Ptet-dcas9, mrcB::msfgfp-mrcB, mrcA::mcherry-mrcA, ΔpbpC, ΔmtgA, ΔmscS* | *Supplementary file 1* |
| Strain, strain background (*E. coli*) | AV93 | This work | 186::*Ptet-dcas9, mrcB::msfgfp-mrcB, mrcA::mcherry-mrcA, ΔpbpC, ΔmtgA, ΔmscS, ΔmscL* | *Supplementary file 1* |
| Strain, strain background (*E. coli*) | AV100 | This work | 186::*Ptet-dcas9, ΔmrcA, ΔmrcB, HK022::Para-msfgfp-mrcB* | *Supplementary file 1* |
| Strain, strain background (*E. coli*) | AV101 | This work | 186::*Ptet-dcas9, ΔmrcA, ΔmrcB, HK022::Para-mcherry-mrcA* | *Supplementary file 1* |
| Strain, strain background (*E. coli*) | AV105 | This work | 186::*Ptet-dcas9, ΔmrcB, mrcA::mcherry-mrcA, ΔpbpC, ΔmtgA, ΔlysA* | *Supplementary file 1* |
| Strain, strain background (*E. coli*) | AV109 | This work | 186::*Ptet-dcas9, mrcB::msfgfp-mrcB, mrcA::mcherry-mrcA, ΔlpoA* | *Supplementary file 1* |
| Strain, strain background (*E. coli*) | AV110 | This work | 186::*Ptet-dcas9, mrcB::msfgfp-mrcB, mrcA::mcherry-mrcA, ΔlpoB* | *Supplementary file 1* |
| Strain, strain background (*E. coli*) | AV124 | This work | 186::*Ptet-dCas9, mrcB::msfgfp-mrcB(E313D)* | *Supplementary file 1* |
| Strain, strain background (*E. coli*) | AV128 | This work | 186::*Ptet-dCas9, mrcB::msfgfp-mrcB(E313D), ΔlpoB* | *Supplementary file 1* |
| strain, strain background (*E. coli*) | AV130 | This work | 186::*Ptet-dCas9, mrcB::msfgfp-mrcB(E313D), mrcA::mcherry-mrcA, ΔlpoB* | *Supplementary file 1* |
| Strain, strain background (*E. coli*) | NO34 | *Ouzounov et al., 2016* | *mreB::mreB-msfgfpsw-kanR* | |
| Strain, strain background (*E. coli*) | B150 | This work | *ΔmrcB* | *Supplementary file 1* |
| Strain, strain background (*E. coli*) | B151 | *van Teeffelen et al., 2011* | FB83, *asd-1* | |
| Strain, strain background (*E. coli*) | B157 | This work | FB83, *asd-1, ΔmrcB* | *Supplementary file 1* |

*Continued on next page*

*Continued*

| Reagent type (species) or resource | Designation | Source or reference | Identifiers | Additional information |
|---|---|---|---|---|
| Strain, strain background (*E. coli*) | B172 | This work | *mreB::mreB-msfgfpsw-kanR* | *Supplementary file 1* |
| Strain, strain background (*E. coli*) | B174 | This work | Δ*mrcB, mreB::mreB-msfgfpsw-kanR* | *Supplementary file 1* |
| Strain, strain background (*E. coli*) | B176 | This work | FB83, *asd-1, mreB::mreB-msfgfpsw-kanR* | *Supplementary file 1* |
| Strain, strain background (*E. coli*) | B178 | This work | FB83, *asd-1,* Δ*mrcB, mreB::mreB-msfgfpsw-kanR* | *Supplementary file 1* |
| Software, algorithm | Trackmate | *Tinevez et al., 2017* | | |
| Software, algorithm | ThunderStorm | *Ovesný et al., 2014* | | |
| Software, algorithm | SpotOn | *Hansen et al., 2018* | | |
| Software, algorithm | Morphometrics | *Ursell et al., 2017* | | |
| software, algorithm | TrackPy | *Allan et al., 2016* | | |
| Software, algorithm | MicroManager | *Edelstein et al., 2010* | | |

## Growth conditions

Cloning and strain preparation were done in Luria-Bertani (LB) medium. In general, every measurement was done in M63 minimal medium with 0.2% glucose, 0.1% casamino-acids and $5 \times 10^{-5}$% thiamine. However, for the experiments described in *Figure 1—figure supplement 2BC* we used 0.5% lactose instead of glucose as a carbon source to allow expression from the $P_{ara}$ promoter. For single-molecule tracking, the concentration of casamino-acids used during the preculture and in the agar pad was only 0.01% to minimize background fluorescence. For the experiments described in *Figure 3—figure supplement 7*, cells were grown in LB medium. The mDAP auxotroph strains were grown in LB supplemented with mDAP (50 µg/ml) and L-homoserine (50 µg/ml, Sigma-Aldrich).

As needed, media were supplemented with kanamycine (50 µg/ml), carbenicillin (100 µg/ml), chloramphenicol (25 µg/ml) or spectinomycin (50 µg/ml), all from Sigma-Aldrich. CRISPR repression is induced with 100 ng/ml of anhydro-tetracycline (Acros Organics). For over-expression of PBP1a or PBP1b from $P_{ara}$, 2 mg/ml of arabinose (Sigma-Aldrich) were added to the medium. For over-expression of LpoB from $P_{lac}$, we added 1 mM of IPTG. The concentration of propidium iodide used to reveal dead cells was 0.4 µM.

Whenever CRISPR knock-down was employed, dCas9 was induced over night so the repressed protein had time to be diluted to steady-state levels. In the morning, the culture was back-diluted 1/500 and grown for at least 3 hr to ensure exponential growth before any experiment. Biological replicates result from independent cultures starting from separate colonies.

## Genetic constructions

All strains used in this study derive from the MG1655. All plasmids constructed in this study were assembled by Gibson assembly. Strains, plasmids, DNA fragments and oligonucleotides are all described in *Supplementary file 1*. Gene deletions were carried out starting from the Keio collection (*Baba et al., 2006*). P1 phage lysate was prepared from the Keio deletion strain, then used to infect the recipient strain and the cells were plated on kanamycine to select for transducers. After each phage P1 transduction, as well as all 'clonetegrations', the kanamycine resistance marker was removed with the flippase-expressing pE-FLP (*St-Pierre et al., 2013*). Integration of RFP-PBP1a and

GFP-PBP1b in the native locus was done using the allelic exchange procedure described in *Vigouroux et al. (2018)*. The CRISPR plasmids are either from the pcrRNA collection described in *Vigouroux et al. (2018)*, or were assembled using the pAV20 double-sgRNA vector (*Dion et al., 2019*). In the later case, complementary oligonucleotide pairs (*Supplementary file 1*) were phosphorylated with T4 PNK in the presence of T4 ligase buffer (New England Biolabs) and then annealed. A mix containing the pAV20 vector, the two pairs of annealed oligos, the BsaI restriction enzyme (New England Biolabs), T4 ligase (New England Biolabs) and ATP was subjected to thermal cycles for digestion, annealing and ligation. The assembly product was subsequently electroporated in DH5 α and the resulting plasmids were sequenced. The 'Ø' control guides, producing no repression, still contain the same 5 bp seed sequence as the GFP- and RFP-targeting guides. This is to account for potential mild 'bad-seed effect' (*Cui et al., 2018*).

## Measurement of optical density and doubling time

Exponential cultures were then transferred to a flat-bottomed 96-microwell plate (Greiner) and optical density at 600 nm was recorded during growth using a microplate reader (Tecan or SAFAS) or, if indicated, using shaking flasks and a spectrophotometer (Eppendorf). A well containing no bacteria was used to subtract the background OD. To calculate the doubling time, we fit an exponential function to the data points corresponding to the exponential phase. To make sure that the exponential phase was properly isolated, we checked that there was no correlation between consecutive residuals after the fit (Durbin-Watson statistic higher than 1).

## Microscopy

For measurements of cell morphology, fluorescence, elasticity and MreB motion, we used an inverted microscope (TI-E, Nikon Inc) equipped with a 100 × phase-contrast objective (CFI PlanApo LambdaDM100 ×1.4 NA, Nikon Inc), a solid-state light source (Spectra X, Lumencor Inc), a multi-band dichroic (69002bs, Chroma Technology Corp.). GFP and RFP fluorescence were measured using excitation filters (560/32 and 485/25 resp.) and emission filters (632/60 and 535/50 resp.). Images were acquired using a sCMOS camera (Orca Flash 4.0, Hamamatsu) with an effective pixel size of 65 nm.

Single particle tracking of GFP-PBP1b was performed in either of two custom-designed fluorescence microscopes, equipped with a custom-built temperature controlled chamber at 29°C or a stage-top incubation chamber (Okolab). Both microscopes were equipped with a 100x TIRF objective (Apo TIRF, 100x, NA 1.49, Nikon), three laser lines: 405 nm (Obis, Coherent), 488 nm (Sapphire, Coherent), 561 nm (Sapphire, Coherent), a dichroic beamsplitter (Di03-R488/561-t3−25 × 36, Semrock) and a laser-line filter (NF561-18, Thorlabs). Shuttering of the 488 nm laser was controlled with an acousto-optic tunable filter (AA Optoelectronics) or with shutters (Uniblitz, LS3 and TS6B, Vincent Associates). Images were acquired with an EMCCD camera (iXon Ultra, Andor). All components were controlled and synchronized using MicroManager (*Edelstein et al., 2010*).

## Measurement of cell morphology and fluorescence

Cells were grown to steady-state exponential phase ($OD_{600} \approx 0.1$) as detailed in 'Growth conditions' and fixed with 4% formaldehyde in phosphate-buffered saline (PBS) for 30 min, except for measuring the fluorescence of cells repressed with sgRNA, which were fixed with 1 mg/ml kanamycine in PBS for 30 min. Fixed cells were transferred to agarose pads (1.5% UltraPure Agarose; Invitrogen) containing PBS and imaged. The Morphometrics package (*Ursell et al., 2017*) was used to find cell contours from phase-contrast images. Cells that are in proximity from each other were excluded using Morphometrics' built-in algorithm. In addition, cells were filtered based on their sharpness in phase-contrast (defined as the variance of gradient magnitude). The fluorescence signal for each image was corrected with $f_{\text{corrected}}(\vec{r}) = (f(\vec{r}) \odot I(\vec{r})) - \langle f(\vec{r}) \odot I(\vec{r}) \odot bg \rangle$ with $f(\vec{r})$ the raw pixel intensities, $I(\vec{r})$ a correction factor for uneven illumination, bg a background mask with no cells and $\odot$ meaning element-wise multiplication. Cell contours were dilated by one pixel to capture all the fluorescence of proteins localized to the membrane, however we found that using different dilatation sizes did not affect the results in a noticeable manner. Intracellular concentration was obtained by integrating the corrected fluorescence intensity inside cell contours, and dividing by cell area. Auto-fluorescence level per pixel inside a cell was calculated by measuring a non-fluorescent strain (LC69) with the

same procedure, and subtracted from cell concentrations. Total regression was used to find the major axis of the cell. The polar regions were detected by setting a threshold on local contour curvature. Cell width was defined as the average distance between the cell contour and this axis, excluding the poles. Cell length was calculated as the maximal distance between contour points projected on the principal cell axis.

## Quantification of PBP1a and PBP1b

The amount of PBP1a and PBP1b following repression by different CRISPR guides was quantified by several methods. First, we measured their expression in AV44 pAV20 GØ-RØ (non-repressed), AV44 pAV20 G14-R20 (strong repression) and LC69 (control strain without fusions) using mass spectrometry. We used Data Independent Acquisitions (DIA) (*Bruderer et al., 2017*) for relative quantification of PBP1a and PBP1b. We also used a targeted proteomics approach, Parallel Reaction Monitoring (PRM) (*Bourmaud et al., 2016*; *Gallien et al., 2012*; *Peterson et al., 2012*), for absolute quantification of PBP1b. We followed the same protocol previously described (*Wollrab et al., 2019*). Peptides used for absolute quantification of PBP1b were based on the FASTA sequence obtained from UniprotKB database and MS evidence of identification. Peptides sequences are LLEATQYR and TVQGASTLTQQLVK (Aqua UltimateHeavy, Thermo Fisher Scientific).

As a confirmation, we used SDS-PAGE with fluorescence detection to compare AV44 pAV20 GØ-RØ to AV44 pAV20 G14-R20. The detailed procedure is described in the following subsection. Finally, for all the strains whose expression level was not quantified by SDS-PAGE or DIA, we used fluorescence microscopy to measure relative expression compared to non-repressed AV44, then used the DIA measurement to obtain PBP1ab expression as a percentage of wild-type level. The expression values obtained from the different methods are shown in *Table 1*.

## Quantification of GFP-PBP1b by SDS-PAGE

A msfGFP-His6 fusion protein was purified to be used as an internal standard for the semi-quantitative msfGFP-PBP1b SDS-PAGE. The msfGFP-6xHis fusion was expressed and purified from a BL21 (DE3) *E. coli* strain. A 10 ml LB preculture containing carbenicillin (100 µg/ml) was inoculated from a freshly transformed colony and grown at 37°C until an $OD_{600} \approx 0.6$. This culture was diluted 1:100 into 500 ml fresh pre-warmed LB containing carbenicillin (100 µg/ml) and grown at 37°C to an $OD_{600} \approx 0.6$. At this time point, the expression was induced by the addition of 1 mM of IPTG and the culture was incubated at 20°C overnight. The next day, the culture was cooled for 15 min at 4°C and the cells were recovered by centrifugation (4000 x g) at 4°C for 15 min. Cell pellets were resuspended in 12.5 ml of lysis buffer (20 mM Tris-HCl pH 8, 100 mM NaCl, 5 mM 2-mercaptoethanol, 20 mM imidazole, 1 mM PMSF) and stored at $-80$°C. Cells were thawed, benzonase (E1014, Millipore) and lysozyme (L6876, Sigma) were added (respectively 500 units and 0.5 mg/ml) and cells were disrupted by sonication on ice. Cell debris and membranes were pelleted by centrifugation at $40,000 \times g$ for 1 hr at 4°C. In parallel, a 2 ml aliquot of Ni-NTA agarose resine slurry (#25214, Thermoscientific), corresponding to a 1 ml beads volume, was equilibrated using 50 ml of buffer with 20 mM of imidazole. The soluble protein extract was incubated with the beads for 1 hr on a wheel at 4°C and loaded on a gravity column. The beads were extensively washed on the column using 50 ml of buffer (20 mM Tris-HCl pH 8, 500 mM NaCl, 5 mM 2-mercaptoethanol, 50 mM imidazole, 10% glycerol). Bound msfGFP-6xHis proteins were eluted in 10 ml buffer (20 mM Tris-HCl pH 8, 500 mM NaCl, 5 mM 2-mercaptoethanol, 120 mM imidazole, 10% glycerol). Fractions of 1 ml were collected and their concentration was estimated using a Bradford-based Protein Assay (Bio-Rad) according to the instructions. The purity of elution fractions was also estimated by loading 5 µl on a 4–20% polyacrylamide gel (Miniprotean TGX, Bio-rad) stained with Coomassie blue and scanned with a Typhoon 9000 FLA imager (GE Healthcare) to detect GFP signal (473 nm laser, excitation wavelength 489 nm, emission 508 nm) (*Figure 1—figure supplement 3A*).

In order to estimate the copy numbers of msfGFP-PBP1b per cell, three independent cell extract preparations of AV44 pAV20-GØ-RØ (non-repressed), AV44 pAV20 G14-R20 and AV51 (ΔPBP1a) pAV20 G14-R20 were analyzed by fluorescence gel-based assay. Cells were grown overnight in LB at 30°C and diluted 1/100 into 40 ml of LB with 100 ng/ml anhydrotetracycline. Three independent cultures, for each strain, were grown at 30°C to an $OD_{600}$ approximately of 0.3 and the colony forming units (cfu) of each culture were determined by plating serial dilutions on LB plates. Cells were

harvested by centrifugation, resuspended in 200 µl of PBS 1x. Cells were disrupted by sonication and protein concentrations were determined using a Bradford-based Protein Assay (5000006, Bio-Rad) according to the instructions. 150 µl of the total cell extract was mixed with 25 µl of Laemmli sample buffer 4X (#1610747, Bio-Rad). Cell extracts were flash-frozen in liquid nitrogen and stored at −80°C. To determine the amount of PBP1b in each of the extracts, normalized amounts of total protein were loaded on 4–20% polyacrylamide gels (Miniprotean TGX, Bio-rad) together with increasing amounts of purified msfGFP-6xHis (same msfGFP used for PBP1b tagging). After migration, the gel was stained with Coomassie blue and scanned for fluorescence as detailed above. A standard curve plotting integrated signal intensity versus protein concentration was generated for the purified msfGFP-6xHis and was used to determine the number of molecules of msfGFP-PBP1b loaded on the gel for each cell extract. The cell number determined for the initial cell cultures were then used to calculate the number of PBP1b molecules per cell.

## mDAP incorporation measurement

This experiment was done with the strains AV84 pAV20-GØ-RØ (non-repressed), AV84 pAV20-G14-R20, AV84 pAV20-G20-RØ (ΔPBP1b) and AV105 pAV20-GØ-RØ (20-Ø) (see *Table 1*). These strains are lacking *lysA* so radio-labeled mDAP is only used for cell-wall synthesis.

Strains were grown to exponential phase and when $OD_{600}$ reached 0.4, $^3$H-labelled mDAP was added for a final activity of 5 µCi/ml. For each time point, 200 µl of culture were transferred to tubes containing 800 µl of boiling 5% SDS. After at least one hour of boiling, the samples were transferred to 0.22 µm GSWP filters. After applying vacuum, the filters were washed twice with 50 ml of hot water. The filters were then moved to 5 ml scintillation vials, treated overnight with 400 µl of 10 mg/ml lysozyme, and dissolved in 5 ml FilterCount cocktail (PerkinElmer) before counting.

The amount of $^3$H-mDAP per cell was calculated by dividing the total counts by the optical density of the culture. The growth rate $\gamma$ was obtained by fitting an exponential function to the $OD_{600}$ values as a function of time. To calculate the incorporation rate $k_{in}$ and turn-over rate $k_{out}$, we fit the data with formula $\frac{^3H-mDAP}{OD_{600}} = \frac{k_{in}}{\gamma + k_{out}}\left(1 - e^{-t*(\gamma + k_{out})t}\right)$ with non-linear least squares optimisation. $k_{out}$ is kept constant across all cultures, assuming there are no difference in turn-over rate.

## UPLC content analysis of peptidoglycan

PG samples were analyzed as described previously in *Alvarez et al. (2016)*, and *Desmarais et al. (2013)*. Briefly, bacterial cultures were harvested, and boiled in 5% SDS for 2 hr. Sacculi were repeatedly washed with MilliQ water by ultracentrifugation (110,000 rpm, 10 min, 20°C) until total removal of the detergent. Samples were treated with 20 µg Proteinase K (1 hr, 37°C) for Braun's lipoprotein removal, and finally treated with muramidase (100 µg/mL) for 16 hr at 37°C. Muramidase digestion was stopped by boiling and coagulated proteins were removed by centrifugation (10 min, 14,000 rpm). For sample reduction, the pH of the supernatants was adjusted to pH 8.5–9.0 with sodium borate buffer and sodium borohydride was added to a final concentration of 10 mg/mL. After incubating for 30 min at room temperature, pH was adjusted to 3.5 with orthophosphoric acid. UPLC analyses of muropeptides were performed on a Waters UPLC system (Waters Corporation) equipped with an ACQUITY UPLC BEH C18 Column, 130 Å, 1.7 µm, 2.1 mm X 150 mm (Waters) and a dual wavelength absorbance detector. Elution of muropeptides was detected at 204 nm. Muropeptides were separated at 45°C using a linear gradient from buffer A (formic acid 0.1% in water) to buffer B (formic acid 0.1% in acetonitrile) in an 18 min run, with a 0.25 mL/min flow. Relative total PG amounts were calculated by comparison of the total intensities of the chromatograms (total area) from three biological replicates normalized to the same $OD_{600}$ and extracted with the same volumes. Muropeptide identity was confirmed by MS/MS analysis, using a Xevo G2-XS QTof system (Waters Corporation). Quantification of muropeptides was based on their relative abundances (relative area of the corresponding peak) normalized to their molar ratio. For the measurement at different levels of PBP1ab (*Figure 2*), the amount was also normalized with respect to the $OD_{600}$ of the culture at the time of harvesting. The percentage of cross-linking was calculated as $\mathrm{Dimers + Trimers} \times 2$.

## Measurement of cell elasticity and osmotic shock resistance

Osmotic shifts were done by replacing a high-osmolarity medium (M63 with 1/10 vol 5 M NaCl) with a low-osmolarity medium (M63 with 1/10 vol of water) for a shock magnitude of 1 osm. In all cases,

cells were grown overnight in high-osmolarity medium then diluted 1/500 and grown at least 3 hr to reach exponential phase.

To perform osmotic shifts while monitoring cellular dimensions, we constructed a tunnel with two strips of double-sided adhesive tape attached to a glass slide and a cover slip. Poly-L-lysine (Sigma-Aldrich) was flushed in the tunnel then washed once with medium. High-osmolarity medium containing a mix of exponentially-growing cells with and without repression of PBP1ab was then flushed in the tunnel. The slide was incubated 15 min for cells to settle. Fresh medium was flushed again to remove unattached cells. Then we took images of GFP and RFP fluorescence, to quantify the total amount of GFP-PBP1b and RFP-PBP1a in each cell. This allowed to distinguish non-repressed cells from repressed ones without ambiguity. We finally recorded phase-contrast images while the low-osmolarity medium was flushed into the tunnel. Cells were tracked using a simple nearest-neighbor algorithm, discarding the cells that went out of focus.

For osmotic shock resistance, cells were prepared in a similar manner, then growth in high-osmolarity was monitored for 6 hr in a plate reader. The plate was then centrifuged 2 min at $2000 \times g$, medium was discarded and low-osmolarity medium was added instead. Growth was then monitored again for 4 hr.

## Measurement of MreB-GFP motion

For measurements of cell boundaries, we focused on cells based on the phase-contrast signal. To track MreB-GFP spots moving at the bottom of the cell, we moved the focal plane 250 nm below the central plane of cells. Images were taken every 2 s for a duration of 120 s.

Images were analyzed using a custom Matlab code as described previously (*Wollrab et al., 2019*). MreB-GFP images are first denoised by using a bandpass filter (bpass function from https://site.physics.georgetown.edu/matlab/code.html) with highpass and lowpass parameters of 100 and 0.5 pixels, respectively. Images were subsequently rescaled by a factor of 5 using spline interpolation to achieve sub-pixel resolution. MreB spots were detected as local maxima inside the cell boundary obtained by segmentation using the Morphometrics package (*Ursell et al., 2017*). MreB spots with an higher intensity than the cell background were considered for tracking. Local maxima were connected to construct raw trajectories based on their distance at consecutive time points (*van Teeffelen et al., 2011*) with a maximal displacement during subsequent time frames of 3 pixels. After generating the tracks, we applied a Gauss filter in time ($\sigma = 1.5$ time steps) in order to decrease spatial noise. Tracks which have more than seven localizations were considered for velocity distributions. Velocity is calculated from single displacement vectors of the smoothened trajectories. Flux is then calculated by summing over all end-to-end distances of smoothened tracks that are longer than 200 nm, normalized by total duration of the movie (2 min for all movies) and total surface area of all cells.

## Single-particle tracking of PBP1b

Prior to imaging, cells were centrifuged for 1 min and transferred to a pre-heated 1% agarose pad (Invitrogen) and covered with a pre-cleaned cover slip. Cover slips were cleaned by bath sonication in a 1M KOH solution for 1 hr at 40°C. For D-cycloserine treatment (1 mM) (*Figure 4D*), cells were treated in liquid culture (M63 minimal medium as indicated above) for 15, and 30 min, respectively, before placing cells on agarose pads containing the same medium and drug for SPT. Time indicates total time of treatment. For recovery from D-cycloserine (*Figure 4E*), cells were treated with D-cycloserine for 30 min (1 mM) in liquid culture, then washed and grown in medium without drug for 3, 4, 15, or 30 min, and then placed on agarose pads.

Images were acquired with exposure time and intervals of 60 ms for a duration of 20 s to 1 min. To distinguish single molecules, this requires a photobleaching phase prior to image acquisition. To that end, the sample was exposed to 488 nm laser in epifluorescence or HILO (highly inclined and laminated optical sheet) mode. Since both modalities resulted in the same fractions of bound molecules, the bleaching or illumination modality did apparently not bias toward either state of molecules. After photobleaching, we either switched to HILO mode or remained in HILO mode for image acquisition. Bleaching time is adjusted according to the level of PBP1b and illumination intensity and was about 2 or 12 s for HILO and epi illumination, respectively. A longer bleaching time of 10 or 25 s was required for GFP-PBP1b overexpression (*Figure 4A*, AV51 without repression).

To determine the bound fraction of GFP-PBP1b, we excluded areas without cells using the bright-field channel and a combination of standard pixel-based morphological operations. PBP1b spots in fluorescence images were identified using the ThunderStorm plug-in for ImageJ (*Ovesný et al., 2014*) with wavelet filtering. The peak detection threshold was equal to the standard deviation of the first wavelet levels of input image (Wave.F1). Sub-pixel resolution was achieved by finding the center of a two-dimensional Gaussian fitted to the intensity profile of each spot. Spots in subsequent frames were then connected using the nearest-neighbor algorithm from TrackPy with a maximum step length of 300 nm (*Allan et al., 2016*). For D-cycloserine treatment experiments, TrackMate (*Tinevez et al., 2017*) was used for tracking with the same maximum step length and allowance for two gaps. Here, we omitted cells segmentation as the number of localizations outside of the cells was very low. To limit tracking mistakes, we discarded the frames where the peak density was too high by only taking the last 15,000 peaks of each movie into account. The displacements were fit using a two-state diffusion model from the SpotOn software package (*Hansen et al., 2018*), allowing to recover the percentage of bound molecules, the peak localization precision and the free molecules' diffusion constant. In the reference strain (strain AV44 with near-WT levels of PBP1a and PBP1b) and in D-cycloserine-treatment/recovery experiments the diffusion constant of the 'bound' molecules was left as a free parameter and found to be compatible with immobilization of these molecules ($D_{bound} < 0.001$ $\mu m_2$/s, top). For the rest of the analysis, we fixed $D_{bound} = 0$.

## Bocillin-labeling of the PBPs

The bocillin-binding assay used to check the absence of non-fluorescent PBP1ab is similar to what is used in *Cho et al. (2016)* and *Kocaoglu et al. (2012)*. We prepared exponentially-growing cells at $OD_{600} \approx 0.4$. We washed 1.8 ml of each culture in PBS, resuspended them in 200 µl PBS and kept cultures on ice. We disrupted cells by sonication (FB120, Fisher Scientific) and centrifuged them for 15 min at 4°C (21,000 g). We subsequently resuspended the pellet corresponding to the membrane fraction in 50 µl PBS containing 15 µM fluorescently labelled Bocillin-FL (Invitrogen). Membranes were incubated at 37°C for 30 min and washed once in 1 ml PBS. We centrifuged the membranes for 15 min (21,000 g) and resuspended them in 50 µl PBS to remove unbound Bocillin-FL. We measured the protein concentration of each sample with a colorimetric assay based on the Bradford method (Bio-Rad) and loaded equal amounts of protein mixed with 4X Laemmli buffer onto a 10% polyacrylamide gel (Miniprotean TGX, Bio-rad). We visualized the labelled proteins with a Typhoon 9000 FLA imager (GE Healthcare) with excitation at 488 nm and emission at 530 nm.

## Acknowledgements

We thank T Bernhardt for providing the pHC942 plasmid, W Vollmer for providing the pBAD33-pbp1b plasmid, Lun Cui for providing strain LC69, Richard Wheeler for assistance with peptidoglycan extraction and chromatography, and Eva Wollrab and Francois Simon for help with single-molecule tracking. This work was supported by the European Research Council (ERC) under the Europe Union's Horizon 2020 research and innovation program [Grant Agreement No. (679980)], the French Government's Investissement d'Avenir program Laboratoire d'Excellence 'Integrative Biology of Emerging Infectious Diseases' (ANR-10-LABX-62-IBEID), the Mairie de Paris 'Emergence(s)' program, and the Volkswagen Foundation. Research in the Cava lab is supported by MIMS, the Knut and Alice Wallenberg Foundation (KAW), the Swedish Research Council and the Kempe Foundation.

## Additional information

### Funding

| Funder | Grant reference number | Author |
|---|---|---|
| H2020 European Research Council | 679980 | Sven van Teeffelen |
| Agence Nationale de la Recherche | ANR-10-LABX-62-IBEID | Antoine Vigouroux<br>David Bikard<br>Sven van Teeffelen |
| Volkswagen Foundation | | Sven van Teeffelen |

| Mairie de Paris | Emergence(s) | Sven van Teeffelen |
| --- | --- | --- |
| H2020 European Research Council | 677823 | David Bikard |
| Knut och Alice Wallenbergs Stiftelse | | Felipe Cava |
| Swedish Research Council | | Felipe Cava |
| Kempe Foundations | | Felipe Cava |
| Laboratory for Molecular Infection Medicine Sweden | | Felipe Cava |

The funders had no role in study design, data collection and interpretation, or the decision to submit the work for publication.

## Author contributions

Antoine Vigouroux, Conceptualization, Data curation, Formal analysis, Validation, Investigation, Visualization, Methodology; Baptiste Cordier, Conceptualization, Validation, Investigation, Methodology; Andrey Aristov, Data curation, Software, Validation, Investigation, Methodology; Laura Alvarez, Data curation, Formal analysis, Investigation; Gizem Özbaykal, Formal analysis, Methodology; Thibault Chaze, Data curation, Formal analysis, Validation, Investigation, Methodology, mass spectrometry; Enno Rainer Oldewurtel, Software, Methodology; Mariette Matondo, Data curation, Formal analysis, Validation, Methodology, mass spectrometry; Felipe Cava, Data curation, Formal analysis, Supervision, Validation, Investigation, UPLC analysis; David Bikard, Supervision, Funding acquisition, Methodology; Sven van Teeffelen, Conceptualization, Supervision, Funding acquisition, Validation, Methodology, Project administration

## Author ORCIDs

Antoine Vigouroux https://orcid.org/0000-0002-8398-5073
Baptiste Cordier https://orcid.org/0000-0002-6042-9787
Sven van Teeffelen https://orcid.org/0000-0002-0877-1294

## Decision letter and Author response

Decision letter https://doi.org/10.7554/eLife.51998.sa1
Author response https://doi.org/10.7554/eLife.51998.sa2

# Additional files

## Supplementary files

• Supplementary file 1. Strains, plasmids, DNA fragments and oligonucleotides used in this study. All gene deletions were done by P1 transduction from the Keio collection (*Baba et al., 2006*). P$_{tet}$-*dcas9* refers to the cassette described in *Cui et al. (2018)* that minimizes the 'bad seed' effect. MG1655 is a gift from Didier Mazel.

• Supplementary file 2. Relative abundance of the different peaks of muramidase-digested peptidoglycan, measured by UPLC. Strains AV84 and AV105 are seen in *Figure 2*. MG1655 with or without D-cycloserine is seen in *Figure 3*.

• Transparent reporting form

## Data availability

All data generated or analysed during this study are included in the manuscript and supporting files or deposited on Dryad (https://doi.org/10.5061/dryad.m37pvmcxq). Source data files have been provided for Figures 1-4.

The following dataset was generated:

| Author(s) | Year | Dataset title | Dataset URL | Database and Identifier |
|---|---|---|---|---|
| van Teeffelen S, Vigouroux A, Cordier B, Aristov A, Oldewurtel ER, Özbaykal G, Chaze T, Matondo M, Bikard D | 2020 | Class-A penicillin binding proteins do not contribute to cell shape but repair cell-wall defects | https://doi.org/10.5061/dryad.m37pvmcxq | Dryad Digital Repository, 10.5061/dryad.m37pvmcxq |

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
