## [Decision Letter]

**Acceptance summary:**

The manuscript from Vigouroux and co-workers details an investigation into the function of class A penicillin-binding proteins (aPBPs) in cell wall repair in *E. coli*. The work is timely because it has recently become apparent that bacteria possess two types of cell wall synthases, the bi-functional aPBPs that possess both peptidoglycan (PG) polymerase and crosslinking activities on a single polypeptide, and the SEDS-Class B PBP (bPBP) complexes where the activities are provided by the two-component synthase. Understanding the relative roles of each of these synthases in overall cell wall assembly has remained an important outstanding question.

This study builds on prior work from Cho et al., 2016 where it was found that aPBPs function semi-autonomously from the elongation/Rod machinery that employs the SEDS-bPBP complex of RodA-PBP2. Here, the authors established a tunable CRISPR/dCas9 system to control the levels of PBP1a and PBP1b, and showed that changing levels of PBP1ab do not influence cell diameter or the amount of newly inserted PG dramatically, nicely confirming previous results in the field in a rigorous and quantitative way. They then go on to provide evidence in support of a model in which aPBPs provide mechanical stiffness to the wall by identifying and fortifying local wall defects, which are likely large cell wall pores sensed by LopB. Overall, the work provides important new findings that advance our understanding of the function of aPBPs in the cell wall assembly. It will be of broad interest to *eLife* readers.

**Decision letter after peer review:**

Thank you for submitting your article "Cell-wall synthases contribute to bacterial cell-envelope integrity by actively repairing defects" for consideration by *eLife*. Your article has been reviewed by three peer reviewers, including Jie Xiao as the Reviewing Editor and Reviewer #1, and the evaluation has been overseen by Anna Akhmanova as the Senior Editor. The following individual involved in review of your submission has agreed to reveal their identity: Tobias Dörr (Reviewer #2).

The reviewers have discussed the reviews with one another and the Reviewing Editor has drafted this decision to help you prepare a revised submission.

Essential revisions:

1) The authors should provide more definitive evidence to show that PBP1b is indeed "actively" repair cell wall defect if this were a major conclusion they would like to claim. The main evidence the authors used to support their conclusion that PBP1b actively repairs cell-wall damage is from the rescuing experiments, where overexpression of PBP1b rescues cells with depleted cell wall precursors. I do not fully understand why this evidence would support an "active" role of PBP1b in damage repair. If the authors use a cell strain that overexpresses the Rod system, will they observe a similar effect? If not, they would have better evidence supporting their conclusion. But if so, should they conclude that the Rod system is also actively repair damages? The drug-washout and PBP1b overexpression experiment did show that PBP1b is critical in cells' recovery from precursor depletion, but PBP1b does not need to in an "active manner" to repair cell defects. They only need to be functional. If to "demonstrate that PBP1b responds to cell wall damage", definitive evidence such as the ones described above, or that showing direct recruitment of aPBPs to areas of damage would be needed. At minimal, the authors should tone down such definitive statements throughout the text (such as in the first and fifth paragraphs of the Discussion).

2) After precursor depletion, the whole cell wall synthesis slows down, so it is hard to argue that there are specific cell wall defects – the model in the field is that PBP1b is activated when there are larger pores in the cell wall so LopB can reach over the pores to activate PBP1b. Should some amidase hypermutants be better to specify these defects, other than use drug-treatment that reduces the whole cell wall synthesis? (Here I am assuming that without mDap or D-Ala-D-Ala Lipid II would not be synthesized or flipped). Another way to address this would be to use HPLC to quantify PG structure after D-cyc or mDAP treatment to show the specific types of damages, and compare the cell wall structures in WT cells and when PBP1b is never induced or always induced. Again, the authors provided evidences that are consistent with the current model, but these evidences are not definitive. The authors should revise the text accordingly so not to claim such definitive conclusions.

3) The title could be revised to be more specific – "Class A PBPs contribute to…". As written, it seems like an obvious statement (of course cell wall synthases, including RodA/PBP2, contribute to cell-envelope integrity; the novelty lies in the way aPBPs do this).

---

## [Author Response]

Essential revisions:1) The authors should provide more definitive evidence to show that PBP1b is indeed "actively" repair cell wall defect if this were a major conclusion they would like to claim. The main evidence the authors used to support their conclusion that PBP1b actively repairs cell-wall damage is from the rescuing experiments, where overexpression of PBP1b rescues cells with depleted cell wall precursors. I do not fully understand why this evidence would support an "active" role of PBP1b in damage repair. If the authors use a cell strain that overexpresses the Rod system, will they observe a similar effect? If not, they would have better evidence supporting their conclusion. But if so, should they conclude that the Rod system is also actively repair damages? The drug-washout and PBP1b overexpression experiment did show that PBP1b is critical in cells' recovery from precursor depletion, but PBP1b does not need to in an "active manner" to repair cell defects. They only need to be functional. If to "demonstrate that PBP1b responds to cell wall damage", definitive evidence such as the ones described above, or that showing direct recruitment of aPBPs to areas of damage would be needed. At minimal, the authors should tone down such definitive statements throughout the text (such as in the first and fifth paragraphs of the Discussion).

We agree with the reviewers that our conclusion about the’ active’ role of PBP1b as a repair enzyme was too strong. We have therefore substantially changed title, Abstract, and manuscript text. In the following we briefly summarize what we now think can be firmly concluded, and which hypothesis is supported by our experiments but not strictly demonstrated.

First, regarding the word ‘active’, we now think that this word is unclear/ambiguous and should be replaced. An enzyme is active simply through its enzymatic activity. If PBP1b is a repair enzyme, as we mean to support through our experiments, it is automatically an active repair enzyme. What we meant to say with the word ‘active’ was to describe the potential propensity of PBP1b to increase activity in response to larger or more mechanical defects. However, we now agree that increased activity cannot be concluded from our experiments.

We think that our work provides evidence for PBP1b being a repair enzyme, i.e., that it inserts PG at locations of local cell-wall damages. This is to be contrasted with the scenario where PBP1b simply prevents the formation of defects by adding more or ‘better’ (in the sense of more sturdy) peptidoglycan. This conclusion is supported by our previous and new experiments.

a) PBP1b expression rapidly reduces the rate of lysis after transient D-cycloserine treatment – within <20 min after washout in both minimal and rich media. In minimal medium, this time constitutes less than 30% of the generation time. Thus, PBP1b is likely recruited to local cell-wall defects, where it then serves as a repair enzyme. We therefore moved the recovery results obtained in minimal media (previously Figure 4—figure supplement 3) into the main figure and moved the results obtained in rich media (LB) into a supplementary figure (Figure 3—figure supplement 3).

Reduction of lysis between 10-25 min after washout is now also demonstrated in single-cell time-lapse microscopy (the new Figure 3E-G, see also Figure 3—figure supplement 4).

D-cycloserine treatment, in turn, reduces the average density of peptidoglycan and thus increases the average pore size in the cell wall. The increase of pore size is supported by new UPLC experiments conducted in collaboration with Felipe Cava (see Figure 3A, B), and by previous 3H-mDAP labeling experiments presented in [Oldewurtel et al., 2019]. To support this claim, we included the results of the UPLC-UV characterization of sacculi after 1 mM D-cycloserine treatment in Figure 3B.

We do not expect that cell-wall defects are initially qualitatively very different from defects that are routinely generated during regular growth. However, these defects are likely responsible for lysis both during and after D-cycloserine treatment, possibly because they grow over time. Our experiments then suggest that these defects are repaired by PBP1b after D-cycloserine washout.

b) Upon limitation of cell-wall precursors through mDAP depletion or through intermediate (lethal but non-saturating) levels of D-cycloserine we observe that cells survive for longer in the presence of PBP1b. Since we make precursor synthesis limiting in these experiments, these experiments suggest that PBP1b is better capable to maintain mechanical stability than the remaining other enzymes, with the same limited resources. This finding supports previous findings of increased sensitivity of the ∆1b mutant to cell-wall-targeting antibiotics.

We conducted new experiments with a PBP1b* mutant isolated by the Bernhardt lab that doesn’t require LpoB to complement a PBP1a deletion [Markovski et al., 2016; Paradis-Bleau et al., r2010]. We use this mutant to demonstrate that PBP1b with LpoB aids recovery from transient cell-wall damage (D-cycloserine treatment) if compared to PBP1b* ∆LpoB (Figure 3—figure supplement 5).

This interpretation is further supported by single-molecule tracking measurements of PBP1b and PBP1b* in a ∆LpoB background. In both cases the bound fraction of molecules is strongly reduced (Figure 4—figure supplement 4).

These observations support the idea that LpoB serves to sense pores in the cell wall and that this sensing mechanism plays a role for D-cycloserine recovery, as previously speculated.

2) After precursor depletion, the whole cell wall synthesis slows down, so it is hard to argue that there are specific cell wall defects – the model in the field is that PBP1b is activated when there are larger pores in the cell wall so LopB can reach over the pores to activate PBP1b. Should some amidase hypermutants be better to specify these defects, other than use drug-treatment that reduces the whole cell wall synthesis? (Here I am assuming that without mDap or D-Ala-D-Ala Lipid II would not be synthesized or flipped). Another way to address this would be to use HPLC to quantify PG structure after D-cyc or mDAP treatment to show the specific types of damages, and compare the cell wall structures in WT cells and when PBP1b is never induced or always induced. Again, the authors provided evidences that are consistent with the current model, but these evidences are not definitive. The authors should revise the text accordingly so not to claim such definitive conclusions.

We realize that our use of the word ‘defect’ is ambiguous. We have therefore included text to explain what we mean in the section about repair:

“Decreased cross-linking likely causes a concomitant accumulation of defects, that is locations with increased pore size, which are ultimately responsible for lysis.” We used this word to refer to sites of possible future lysis. These might be cell-wall pores of increased size. These defects can in principle be generated during regular growth (where they are rapidly repaired) or during the arrest of cell-wall insertion.

As described in more detail in response to the previous point, we now demonstrated more clearly that pore size is indeed increased. We therefore think that D-cycloserine treatment is well suited to introduce defects in the cell wall and to probe the capacity of PBP1b to repair these otherwise lethal defects.

We agree that our experiments do not provide proof of a particular mechanism of PBP1b, as already discussed above. We have therefore revised Abstract, Introduction, Results, and Discussion.

3) The title could be revised to be more specific – "Class A PBPs contribute to…". As written, it seems like an obvious statement (of course cell wall synthases, including RodA/PBP2, contribute to cell-envelope integrity; the novelty lies in the way aPBPs do this).

We agree with the reviewers. The new title is ‘Class-A penicillin binding proteins do not contribute to cell shape but repair cell-wall defects’.